# Stable coresets: Unleashing the power of uniform sampling

**Amir Carmel**[*]
Weizmann Institute of Science
amir6423@gmail.com

**Robert Krauthgamer**[†]
Weizmann Institute of Science
robert.krauthgamer@weizmann.ac.il

## Abstract

Uniform sampling is a highly efficient method for data summarization. However, its effectiveness in producing coresets for clustering problems is not yet well understood, primarily because it generally does not yield a strong coreset, which is the prevailing notion in the literature. We formulate *stable coresets*, a notion that is intermediate between the standard notions of weak and strong coresets, and effectively combines the broad applicability of strong coresets with highly efficient constructions, through uniform sampling, of weak coresets. Our main result is that a uniform sample of size $O(\epsilon^{-2} \log d)$ yields, with high constant probability, a stable coreset for 1-median in $\mathbb{R}^d$ under the $\ell_1$ metric. We then leverage the powerful properties of stable coresets to easily derive new coreset constructions, all through uniform sampling, for $\ell_1$ and related metrics, such as Kendall-tau and Jaccard. We also show applications to fair rank aggregation and to approximation algorithms for $k$-median problem in these metric spaces. Our experiments validate the benefits of stable coresets in practice, in terms of both construction time and approximation quality.

## 1 Introduction

Clustering is a fundamental problem in data analysis, machine learning, and optimization, facilitating various downstream tasks such as classification, anomaly detection, and efficient retrieval. In general, the input is a set of points in a metric space, and the objective is to partition this set into disjoint clusters, each sharing a high degree of similarity. In center-based clustering, the goal is to further assign a representative point, called center, to each cluster. The famous $k$-median problem, where the number of clusters is denoted by $k$, seeks to minimize the sum of distances from each input point to its assigned center.

In the era of big data, clustering often involves huge datasets, where direct processing is computationally prohibitive. This challenge has given rise to the sketch-and-solve paradigm, which employs a summarization step prior to the desired computational task, i.e., the data is first preprocessed into a compact summary, ideally of size that is independent of the original dataset size, and then the desired algorithms for learning or optimization are applied only to this smaller summary. This approach significantly reduces computational resources, such as running time, memory, and communication, but requires balancing between the summary's size and its information loss.

A coreset represents a concrete formalization of data summarization, and is typically defined as a small subset of the input that provably captures the relevant geometric properties for a specific objective function. Over the past two decades, coresets have been extensively studied and successfully applied to a wide range of problems (see the surveys (Feldman, 2020; Phillips, 2017) and references therein), and two fundamental types of coresets have dominated the literature. A *weak coreset*, which aligns with the sketch-and-solve paradigm, ensures that an optimal or near-optimal solution computed for the coreset is near-optimal also for the original dataset, without necessarily preserving the objective value itself. In contrast, a *strong coreset* provides a more comprehensive guarantee by

---

[*]Part of this work was completed while the author was affiliated with Pennsylvania State University.

[†]The Harry Weinrebe Professorial Chair of Computer Science. Work partially supported by the Israel Science Foundation grant #1336/23.

preserving the objective value, up to small error, for all potential solutions, i.e., all centers in the metric space. This notion is indeed stronger and has broader range of applications. In particular, it exhibits a powerful property: if a metric space $\mathcal{X}$ admits a strong coreset (meaning that every instance in $\mathcal{X}$ admits a small coreset), then every submetric $\mathcal{X}' \subseteq \mathcal{X}$ also admits a strong coreset. It follows that coreset results for one metric space $\mathcal{X}$ extend to every metric that can be embedded isometrically in $\mathcal{X}$. A concrete example here is the Kendall-tau metric on rankings, which embeds into $\ell_1$.

This property regarding submetrics is very useful also when the optimal solution is constrained to meet specific criteria. One such example is the problem of fair rank consensus, which asks to aggregate multiple rankings into a single ranking, viewed as a center point, that satisfies certain fairness constraints among the candidates being ranked (Chakraborty et al., 2022; Patro et al., 2022; Pitoura et al., 2022). Another example emerges in biological research, where molecular data needs to be aggregated while satisfying structural properties that are critical for maintaining molecular stability (Tian et al., 2021; Zeng et al., 2023). Furthermore, constraints can be modified over time, e.g., to reflect knowledge or demands, effectively restricting the center points each time to a different submetric.

However, the advantages of strong coresets come with nontrivial computational challenges, as their construction algorithms must inevitably read the entire dataset.[1] Can we construct coresets in *sublinear time*? Weak coresets demonstrate that this is possible, as they can sometimes be constructed through uniform sampling (Huang et al., 2023a;b; Marom & Feldman, 2019), a highly efficient method that is easy to implement in streaming and distributed settings, and is well-known to be extremely useful in practice. However, uniform sampling cannot reliably capture small clusters, which motivates us to study the fundamental case $k = 1$, where uniform sampling can succeed without additional assumptions about the dataset. This case arises naturally in many practical scenarios and serves as an important building block for the general problem.

We formulate *stable coresets*, a notion that captures key properties of strong coresets while avoiding many of their computational pitfalls. More precisely, they are intermediate between strong and weak coresets, and effectively provide a sweet spot between these two standard notions. We investigate stable coresets within the framework of the 1-median problem under the $\ell_1$ metric. This metric emerges naturally in data analysis, particularly in high-dimensional settings, and also serves as a unifying framework for understanding numerous distance metrics. Indeed, many important metrics–including Hamming, Kendall-tau, Jaccard, tree metrics, various graph-based distances, as well as $\ell_2$–can be embedded into $\mathbb{R}^d$ with the $\ell_1$ metric either isometrically or with low distortion. Yet another application is in computational biology, where the genome median problem seeks to find a consensus genome that minimizes evolutionary distance to a set of input genomes, often using metrics that embed in $\ell_1$ space. The upshot is that results for stable coresets in $\ell_1$ immediately imply new coreset constructions also for these embedded metrics, in contrast to weak coresets, which would require a separate analysis for each individual metric.

## 1.1 PROBLEM SETUP AND DEFINITIONS

In $k$-median, the input is a finite set of points $P$ in the metric space $(\mathcal{X}, \text{dist})$ and the goal is to find a set of $k$ centers $C \subseteq \mathcal{X}$ that minimizes the objective function

$$\text{cost}(C, P) := \sum_{p \in P} \min_{c \in C} \text{dist}(c, p).$$

We focus on the case $k = 1$. For this single-center case, we denote an arbitrary optimal center (which minimizes the objective) by $c^P \in \mathcal{X}$, and the optimal value by $\text{opt}(P) := \text{cost}(c^P, P)$.

We proceed to define formally the three coreset variants discussed above.[2] In a weak coreset, the main idea is that solving the coreset instance optimally, or even approximately, yields an approximately optimal solution also for the original instance. Our definition below uses parameters $\epsilon$ and $\eta$ to create

---

[1]Consider $k = 1$ and a dataset where all points are densely clustered, except for one "outlier" point extremely far away. While this outlier hardly affects the optimal center, its impact on the objective value might be significant, and thus it must be examined and even included in every strong coreset.

[2]Although coresets are traditionally defined as weighted subsets, we present our definitions without weights for sake of clarity, as our focus is on coresets obtained through uniform sampling.

a tunable tradeoff in the approximation guarantee, although most literature restricts attention to a single parameter by setting $\eta = O(\epsilon)$ or alternatively $\epsilon = 0$.

**Definition 1.1** (Weak Coreset). A subset $Q \subseteq P$ is a *weak $(\epsilon, \eta)$-coreset* for a 1-median instance $P \subseteq \mathcal{X}$ if

$$\forall c \in \mathcal{X}, \qquad \mathrm{cost}(c, Q) \le (1 + \epsilon) \operatorname{opt}(Q) \;\to\; \mathrm{cost}(c, P) \le (1 + \eta) \operatorname{opt}(P). \tag{1}$$

A strong coreset provides a more comprehensive guarantee by ensuring that the objective value is preserved for every possible center point in the metric space.

**Definition 1.2** (Strong Coreset). A subset $Q \subseteq P$ is a *strong $\epsilon$-coreset* for a 1-median instance $P \subseteq \mathcal{X}$ if

$$\forall c \in \mathcal{X}, \qquad \mathrm{cost}(c, Q) \in (1 \pm \epsilon) \mathrm{cost}(c, P). \tag{2}$$

A stable coreset imposes geometric constraints on all points in the metric space, similarly to a strong coreset, but with a comparative structure like that of a weak coreset.

**Definition 1.3** (Stable Coreset). A subset $Q \subseteq P$ is a *stable $(\epsilon, \eta)$-coreset* for a 1-median instance $P \subseteq \mathcal{X}$ if

$$\forall c_1, c_2 \in \mathcal{X}, \qquad \mathrm{cost}(c_1, Q) \le (1 + \epsilon) \mathrm{cost}(c_2, Q) \;\to\; \mathrm{cost}(c_1, P) \le (1 + \eta) \mathrm{cost}(c_2, P). \tag{3}$$

A key difference between strong and stable coresets is that the former preserve the actual cost of every center, while the latter preserve only the relative order of the costs across different centers. For illustration, consider a dataset whose points are clustered together except for one distant "outlier". A strong coreset must include this outlier to preserve its large contributions to the cost, while a stable coreset need not. This weaker requirement is crucial for uniform sampling to work effectively, and reveals a natural compatibility between stable coresets and uniform sampling.

While not formalized as a coreset notion, the principle underlying (3) and its compatibility with uniform sampling were first used by Indyk (1999; 2001), to compare the costs of two centers in the context of 1-median with discrete centers (i.e., the center must be one of the dataset points). For finite $\mathcal{X}$, Indyk's analysis would yield a stable $(0, \epsilon)$-coreset of size $O(\epsilon^{-2} \log |\mathcal{X}|)$ (details in Appendix A.1).

We establish some basic properties of these definitions in Section 2. In particular, there is a strict hierarchy: every strong coreset is also a stable coreset, and every stable coreset is also a weak coreset, however the opposite direction is not true in general. In addition, the guarantees of a stable coreset transfer to every submetric, and thus also to any isometrically embedded metric, which is valuable for analyzing discrete metric spaces that embed into $\ell_1$.

## 1.2 OUR CONTRIBUTION

While uniform sampling offers extensive practical advantages, it is often viewed as a heuristic method for constructing coresets, due to limited theoretical foundations. We focus on the case $k = 1$, as extending to $k > 1$ requires additional structural and algorithmic assumptions that we avoided for theoretical clarity. Our main theorem shows that uniform sampling yields stable coresets in a rather broad setting, namely, in $\ell_1$ and thus also in every metric that embeds into $\ell_1$. Our proof has two parts: we first develop a framework for constructing stable coresets that is applicable to all metric spaces (in Section 3), and then we instantiate this framework with $\ell_1$-specific analysis (in Section 4).

**Theorem 1.4.** *Let $P \subset \mathbb{R}^d$ be finite and let $\epsilon \in (0, \frac{1}{5})$. Then, a uniform sample of size $O(\epsilon^{-2} \log d)$ from $P$ is a stable $(\epsilon/6, 4\epsilon)$-coreset for 1-median in $\ell_1^d$ with probability at least $4/5$.*

Prior work on coresets constructed through uniform sampling works in restricted settings. A weak $(0, \epsilon)$-coreset in $\ell_1$ is known from (Danos, 2021), however using it would require solving the problem optimally on the coreset. A weak $(\epsilon, O(\epsilon))$-coreset in $\ell_2$ is known from (Huang et al., 2023a), however it offers much less generality than $\ell_1$ (recall $\ell_2$ embeds in $\ell_1$ with small distortion but not in the opposite direction). There are also weak coresets in doubling metrics (Ackermann et al., 2010; Munteanu & Schwiegelshohn, 2018; Huang et al., 2023a), which include $\ell_1$ and $\ell_2$ spaces, however they are useful only when $\mathcal{X}$ is low-dimensional. Most importantly, these are all weak coresets and need not extend to submetrics. A more comprehensive list of previous results appears in Section 1.4.

Additionally, our approach bridges an important gap – stable coresets provide almost as powerful guarantees as strong coresets, while maintaining the simplicity and efficiency of uniform sampling. It therefore establishes rigorously the broad range of applicability of uniform sampling. We further conjecture that our bound can be improved to be dimension-independent, and our empirical evidence supports this direction.

By utilizing Theorem 1.4 we can establish additional significant results (Section 5). First, we explore implications to metric spaces that embed into $\ell_1$, either isometrically or with small distortion, obtaining the first coresets based on uniform sampling for important metric spaces, including Kendall-tau and Jaccard, as well as new bounds for $\ell_2$. Second, building upon our coreset constructions for 1-median, we derive approximation algorithms for the more general problem of $k$-median, across all the aforementioned metric spaces. Furthermore, we apply our framework to show that in certain scenarios, uniform sampling actually produces strong coresets, which in turn can speed up $O(1)$-approximation algorithms.

Finally, we validate experimentally the performance of our approach in different settings and for various datasets (Section 6). For instance, we show that a uniform sample achieves error rates that are comparable to computationally expensive importance sampling techniques. We also show that when applied to 1-median in the Kendall-tau metric, our coresets effectively preserve the performance of practical heuristic algorithms (which are employed because this optimization problem is NP-hard). We further validate that our coresets for Kendall-tau are effective, i.e., maintain the solution quality, even when constraints such as fairness requirements are imposed on the solution.

### 1.3 TECHNICAL OVERVIEW

We now outline the proof of our main theorem, which consists of two parts, a general framework for arbitrary metric spaces (Section 3) and its concrete application to the $\ell_1$ metric (Section 4).

Our framework establishes a key condition for a subset $Q \subset P$ to be a stable coreset for $P$, called *relative cost-difference approximation*. It asserts that for every potential center in the metric space, the difference between its cost and the median's cost remains approximately the same when measured relative to the input $P$ or to subset $Q$, see (4). This condition is not sufficient by itself and we need another condition, which is rather simple and holds with constant probability for a uniform sample.

To prove that a uniform sample in $\ell_1$ satisfies this condition, we leverage $\epsilon$-approximations, a technique from PAC learning that is tightly connected to the range-counting problem in computational geometry. Li, Long, and Srinivasan (Li et al., 2001) provided tight bounds for this problem, which we apply in our analysis. While $\epsilon$-approximations support range-counting queries (i.e., ensures that the proportion of points in any range is preserved), we show through careful analysis that in $\ell_1$ metrics, $\epsilon$-approximations for axis-aligned half spaces directly translate to preserving the relative cost structure across the entire metric space. We further establish that the query family of axis-aligned half spaces has VC dimension that is logarithmic in the dimension of the underlying space.

### 1.4 RELATED WORK

In $\ell_p$ metric spaces, the $k$-median problem for general $k$ is APX-hard (Guruswami & Indyk, 2003; Trevisan, 2000), with some recent advances about its inapproximability (Cohen-Addad et al., 2022). In a metric space of bounded doubling dimension $D$, this problem admits a polynomial-time approximation scheme (PTAS), namely, $(1 + \epsilon)$-approximation that runs in time $\tilde{O}(2^{(\frac{1}{\epsilon})^{O(D^2)}} n)$ (Cohen-Addad et al., 2021a). Clearly this approach is only practical when the doubling dimension is very small.

Coresets for $k$-median have been researched extensively over the years, with particular emphasis on strong coresets in Euclidean space, see (Feldman, 2020; Munteanu & Schwiegelshohn, 2018) for surveys and (Cohen-Addad et al., 2025; Huang et al., 2025; 2024) for the latest results. Uniform-sampling-based coreset constructions originated in (Chen, 2009), which proposed partitioning the metric space into "rings" and sampling uniformly from each part. This approach was further improved in (Braverman et al., 2022; Cohen-Addad et al., 2021b), and while it yields strong coresets, the overall sampling distribution is non-uniform, and thus the running time is not sublinear.

To enable truly uniform sampling, we must restrict our attention to weaker coresets and the case $k = 1$. Uniform sampling was shown to yield weak $(0, \epsilon)$-coresets for 1-median in Euclidean space

in (Ackermann et al., 2010; Munteanu & Schwiegelshohn, 2018; Danos, 2021), and these bounds were improved by Huang et al. (2023a) to weak $(\epsilon, O(\epsilon))$-coresets of size $\tilde{O}(\frac{1}{\epsilon^3})$, alongside additional results for spaces of bounded doubling dimension and graphs of bounded treewidth (and also an extension to general $k$ under additional assumptions about the dataset). For 1-median in $\ell_1$, a uniform sample of size $\tilde{O}(\frac{1}{\epsilon^2})$ produces a weak $(0, \epsilon)$-coreset (Danos, 2021).

In comparison, strong coresets for $k$-median in $\ell_1$ of size $\text{poly}(k/\epsilon)$ follow implicitly by Jiang et al. (2024), because $\ell_1$ is contained in $\ell_2$-squared, however, constructing such coresets requires at least linear time.

Coresets for 1-center (aka Minimum Enclosing Ball) in $\ell_1$ and in related metrics were studied by Carmel et al. (2025). It was shown that for both strong and weak coresets, the coreset size must depend on the dimension (in contrast to 1-median).

## 2 PRELIMINARIES

We begin by showing that every strong coreset is also a stable coreset, and every stable coreset is a weak coreset, forming a clear hierarchy among these definitions.

**Proposition 2.1.** *Let $(\mathcal{X}, \text{dist})$ be a metric space and let $P \subseteq \mathcal{X}$ be a 1-median instance.*

*(a). Every stable $(\epsilon, \eta)$-coreset of $P$ is also a weak $(\epsilon, \eta)$-coreset.*

*(b). Every strong $\epsilon$-coreset of $P$, for $\epsilon \in (0, \frac{1}{5})$, is also a stable $(\epsilon, 4\epsilon)$-coreset.*

We next describe how stable coreset properties are preserved for submetrics through isometric embeddings. An *isometric embedding* between metric spaces $(\mathcal{X}_1, \text{dist}_1)$ and $(\mathcal{X}_2, \text{dist}_2)$ is a mapping $f : \mathcal{X}_1 \to \mathcal{X}_2$ such that for every $x, y \in \mathcal{X}_1$, we have $\text{dist}_1(x, y) = \text{dist}_2(f(x), f(y))$. The following fact is immediate.

**Fact 2.2.** *Let $f : \mathcal{X}_1 \to \mathcal{X}_2$ be an isometric embedding between metric spaces $(\mathcal{X}_1, \text{dist}_1)$ and $(\mathcal{X}_2, \text{dist}_2)$. Then,*

*(a). $f$ is injective; and*

*(b). for every $P \subseteq \mathcal{X}_1$ and $c \in \mathcal{X}_1$, $\text{cost}(c, P) = \text{cost}(f(c), f(P))$.*

Observe that when $f$ is injective, every subset of $f(P)$ can be written as $f(Q)$ for some $Q \subseteq P$.

**Proposition 2.3.** *Let $f : \mathcal{X}_1 \to \mathcal{X}_2$ be an isometric embedding between metric spaces $(\mathcal{X}_1, \text{dist}_1)$ and $(\mathcal{X}_2, \text{dist}_2)$. For every $Q \subseteq P \subseteq \mathcal{X}_1$, if $f(Q)$ is a stable $(\epsilon, \eta)$-coreset of $f(P)$ in $\mathcal{X}_2$, then $Q$ is a stable $(\epsilon, \eta)$-coreset of $P$ in $\mathcal{X}_1$.*

## 3 A FRAMEWORK FOR STABLE CORESETS

We develop a general framework for proving that subsets $Q \subseteq P$ in arbitrary metric spaces $\mathcal{X}$ are stable coresets, with detailed proofs provided in Appendix B. We apply this framework to $\ell_1$ spaces in Section 4, and believe that it will lead to more results in the future.

We use the notation from Section 1.1, and define also the normalized terms $\overline{\text{cost}}(x, P) := \frac{1}{|P|} \text{cost}(x, P)$ and $\overline{\text{opt}}(P) := \frac{1}{|P|} \text{opt}(P)$. Denoting by $\mu \in \mathcal{X}$ an optimal median point for $P$, we say that $Q$ is an $\epsilon$-*relative cost-difference approximation ($\epsilon$-RCDA)* of $P$ in $\mathcal{X}$ if

$$\forall x \in \mathcal{X}, \qquad \left| \left[ \overline{\text{cost}}(x, P) - \overline{\text{cost}}(\mu, P) \right] - \left[ \overline{\text{cost}}(x, Q) - \overline{\text{cost}}(\mu, Q) \right] \right| \leq \epsilon \cdot \overline{\text{cost}}(x, P). \quad (4)$$

Intuitively, this condition requires that $Q$ preserves the gap in cost between every potential center and a reference point, ensuring that the ranking of centers remains approximately the same whether evaluated on the original set $P$ or the coreset $Q$. We remark that it is not crucial for this definition to have $\mu$ be a median point, and it can be substituted by any fixed point in the metric space, up to constant-factor loss in $\epsilon$. We now use this condition to establish that $Q$ is a stable coreset.

**Theorem 3.1.** *Let $P \subseteq \mathcal{X}$ and $\epsilon \in (0, \frac{1}{5})$, and suppose $\overline{\text{cost}}(\mu, Q) \leq c \cdot \overline{\text{cost}}(\mu, P)$ for some $c \geq 1$. If $Q \subseteq P$ is an $\epsilon$-RCDA of $P$ in $\mathcal{X}$ then $Q$ is a $(\frac{\epsilon}{c}, 4\epsilon)$-stable coreset of $P$.*

## 4 STABLE CORESETS IN $\ell_1$ THROUGH UNIFORM SAMPLING

In this section we prove the following theorem (some proofs appear in Appendix C). As usual, $\ell_1^d$ denotes the metric space $(\mathbb{R}^d, \|\cdot\|_1)$.

**Theorem 1.4.** *Let $P \subset \mathbb{R}^d$ be finite and let $\epsilon \in (0, \frac{1}{5})$. Then, a uniform sample of size $O(\epsilon^{-2} \log d)$ from $P$ is a stable $(\epsilon/6, 4\epsilon)$-coreset for $1$-median in $\ell_1^d$ with probability at least $4/5$.*

For a point $x \in \mathbb{R}^d$, we denote its $i$-th coordinate by $x[i]$. Let $\mathcal{T} := \{\tau_{i,r} : i \in [d], r \in \mathbb{R}\}$ denote the class of threshold functions with $\tau_{i,r}(x) := \mathbb{1}_{\{x[i] \leq r\}}$ for $i \in [d]$ and $r \in \mathbb{R}$.

**Definition 4.1** (VC dimension (Vapnik & Chervonenkis, 1971))**.** Let $\mathcal{F}$ be a class of functions from $\mathcal{X}$ to $\{0, 1\}$. The *growth function* of $\mathcal{F}$ is defined as

$$\forall \text{ integer } m \geq 1, \qquad \mathbf{G}_{\mathcal{F}}(m) := \max_{x_0, \ldots, x_{m-1} \in \mathcal{X}} |\{(f(x_0), \ldots, f(x_{m-1})) : f \in \mathcal{F}\}|,$$

and $\mathbf{G}_{\mathcal{F}}(0) := 1$. The *VC dimension* of $\mathcal{F}$, denoted by $\mathrm{VCdim}(\mathcal{F})$, is the largest $m \geq 0$ such that $\mathbf{G}_{\mathcal{F}}(m) = 2^m$. Furthermore, a set $\{x_0, \ldots, x_{m-1}\}$ such that $|\{(f(x_0), \ldots, f(x_{m-1})) : f \in \mathcal{F}\}| = 2^m$ is called a *shattering set*.

We now bound the VC dimension of the class of threshold functions $\mathcal{T}$, showing that it is essentially logarithmic in the dimension $d$. While tight bounds appear in (Gey, 2018), we provide a shorter proof for completeness.

**Proposition 4.2.** $\lfloor \log d \rfloor \leq \mathrm{VCdim}(\mathcal{T}) \leq 2 \log d$.

For a given $P$, define the empirical distribution function for the $i$-th coordinate by $\mathrm{edf}_P(i, r) := \frac{1}{|P|} \sum_{p \in P} \tau_{i,r}(p) = \frac{1}{|P|} |\{p \in P : p[i] \leq r\}|$. When $d = 1$, we slightly abuse notation and omit the parameter $i$. We use this notation to define $\epsilon$-approximation for $P$.

**Definition 4.3.** Let $P \subset \mathbb{R}^d$ be finite and let $\epsilon \in (0, 1)$. A subset $Q \subseteq P$ is an *$\epsilon$-approximation* for $P$ if

$$\forall i \in [d], \forall r \in \mathbb{R}, \qquad |\mathrm{edf}_Q(i, r) - \mathrm{edf}_P(i, r)| \leq \epsilon.$$

Using a theorem established by Li et al. (2001), we can bound the size of such $\epsilon$-approximation (see also (Har-Peled & Sharir, 2011)).

**Theorem 4.4** (Li et al. (2001))**.** *Let $\mathcal{F}$ be a class of function from $P$ to $\{0, 1\}$, with finite VC dimension, and let $\mathcal{D}$ be some probability distribution over $P$. Then, with probability at least $1 - \delta$, a random sample $Q \sim D$ of size $O\left(\epsilon^{-2}(\mathrm{VCdim}(\mathcal{F}) + \log \frac{1}{\delta})\right)$ satisfies*

$$\forall f \in \mathcal{F}, \qquad \left| \frac{1}{|Q|} \sum_{x \in Q} f(x) - \mathbb{E}_{x \sim D}[f(x)] \right| \leq \epsilon.$$

We now apply Theorem 4.4, taking $\mathcal{F} = \mathcal{T}$ and $\mathcal{D}$ as the uniform distribution over $P$, and use the VC dimension bound from Proposition 4.2.

**Corollary 4.5.** *Let $P \subset \mathbb{R}^d$ be finite and let $\epsilon \in (0, 1)$. With probability at least $1 - \delta$, a uniform sample $Q \subseteq P$ of size $O(\epsilon^{-2} \log \frac{d}{\delta})$ is an $\epsilon$-approximation for $P$.*

We now turn to showing how the above machinery can be applied to the $1$-median problem under the $\ell_1$ metric. Our main technical lemma, shows that an $\epsilon$-approximation subset is also $O(\epsilon)$-RCDA.

**Lemma 4.6.** *Let $\epsilon \in (0, 1)$ and let $P$ be a finite set in $\ell_1^d$. If $Q$ is an $\epsilon$-approximation of $P$, then $Q$ is a $20\epsilon$-RCDA of $P$.*

*Proof of Theorem 1.4.* By Corollary 4.5 and Lemma 4.6, a uniform sample $Q$ of size $O(\epsilon^{-2} \log d)$ yields an $\epsilon$-RCDA of $P$ with large constant probability. Observe that

$$\mathbb{E}[\overline{\mathrm{cost}}(\mu, Q)] = \mathbb{E}\left[ \frac{1}{|Q|} \sum_{q \in Q} \|\mu - q\|_1 \right] = \frac{1}{|P|} \sum_{p \in P} \|\mu - p\|_1 = \overline{\mathrm{cost}}(\mu, P),$$

and thus by Markov's inequality, $\Pr[\overline{\mathrm{cost}}(\mu, Q) \geq 6 \overline{\mathrm{cost}}(\mu, P)] \leq \frac{1}{6}$. By a union bound, with probability at least $\frac{4}{5}$, we have both that $Q$ is an $\epsilon$-RCDA of $P$ and that $\overline{\mathrm{cost}}(\mu, Q) < 6 \overline{\mathrm{cost}}(\mu, P)$. To complete the proof of Theorem 1.4, we now apply our framework, namely, Theorem 3.1. $\qquad \square$

## 5 APPLICATIONS

Several applications of Theorem 1.4 follow immediately from known isometric embeddings into $\ell_1$. In particular, Hamming distance, Kendall-tau, Spearman-footrule, and certain graph-based metrics, such as tree metrics, can all be embedded isometrically into $\ell_1$, allowing our coreset results to transfer directly to these spaces, see e.g. Corollary D.1. Another application that arises in computational biology is the genome-median problem, where the goal is to find a consensus genome that minimizes the total evolutionary distance to a set of input genomes. The breakpoint distance is a common metric for genomic comparison that can sometimes (i.e., under some restrictions) be embedded isometrically in $\ell_1$, with only a quadratic increase in the dimension (Tannier et al., 2009), making our approach applicable. Below, we address the Jaccard metric separately, since its isometric embedding requires a high dimension; however, low-distortion embeddings can be easily employed instead.

**Near-isometric embeddings.** We extend our results to metrics that embed into $\ell_1$ with distortion close to 1, showing they admit stable coresets with parameters adjusted according to the distortion. As usual, an *embedding* between metric spaces $(\mathcal{X}_1, \mathrm{dist}_1)$ and $(\mathcal{X}_2, \mathrm{dist}_2)$ is a map $f : \mathcal{X}_1 \to \mathcal{X}_2$. We say that it has *distortion* $D^2 \geq 1$, if there exists $r > 0$ (scaling factor) such that

$$\forall x, y \in \mathcal{X}_1, \qquad \frac{1}{D} \cdot \mathrm{dist}_2(f(x), f(y)) \leq r \cdot \mathrm{dist}_1(x, y) \leq D \cdot \mathrm{dist}_2(f(x), f(y)). \qquad (5)$$

One can often assume that $r = 1$ by scaling $f$, e.g., when the target $\mathcal{X}_2$ is a normed space. We mostly use the case $D = 1 + \zeta$ for $\zeta \in (0, 1)$, and then the distortion is $D^2 = 1 + O(\zeta)$.

**Proposition 5.1.** *Let* $f : \mathcal{X}_1 \to \mathcal{X}_2$ *be an embedding between metric spaces* $(\mathcal{X}_1, \mathrm{dist}_1)$ *and* $(\mathcal{X}_2, \mathrm{dist}_2)$ *with distortion* $D$ *and scaling factor* $r$. *For every* $Q \subseteq P \subseteq \mathcal{X}_1$, *if* $f(Q)$ *is a stable* $(\epsilon, \eta)$-*coreset of* $f(P)$ *in* $\mathcal{X}_2$ *for some* $\epsilon, \eta > 0$, *and the values* $\epsilon' := (1 + \epsilon)/D^2 - 1$ *and* $\eta' := D^2(1 + \eta) - 1$ *are positive, then* $Q$ *is a stable* $(\epsilon', \eta')$-*coreset of* $P$ *in* $\mathcal{X}_1$.

This proposition extends our results in Theorem 1.4 to metric spaces that can be embedded into $\ell_1$ with small distortion. For example, our results extend to the Euclidean metric using Dvoretzky's Theorem (Gordon, 1988; Schechtman, 2006), yielding stable coresets of size $O(\epsilon^{-2} \log(d/\epsilon))$ for 1-median in $\ell_2^d$, see Appendix D.2. We remark that it suffices to have the distortion guarantee (5) only for pairs that involve a point from $P$, which is known in the literature as *terminal embedding*, see Appendix D.1.

**Corollary 5.2.** *Let* $(\mathcal{X}, \mathrm{dist})$ *be a metric space that embeds in* $(\mathbb{R}^d, \|\cdot\|_1)$ *with distortion* $1 + \frac{\epsilon}{3}$ *for* $\epsilon \in (0, \frac{1}{10})$. *Then a uniform sample of size* $O(\epsilon^{-2} \log d)$ *from a finite* $P \subseteq \mathcal{X}$ *is a stable* $(\epsilon, O(\epsilon))$-*coreset for* 1-*median in* $\mathcal{X}$ *with probability at least* $\frac{4}{5}$.

**Stable coresets in Jaccard metric.** Consider the Jaccard metric over $d$ elements, i.e., over ground set $[d]$ without loss of generality. It follows immediately from (Broder et al., 1998) that the Jaccard metric embeds with distortion $1 + \zeta$ into $\ell_1$ space of dimension $O(\zeta^{-2} d^3)$. Thus, Corollary 5.2 implies coresets for the Jaccard metric, as follows.

**Corollary 5.3.** *Let* $P \subseteq 2^{[d]}$ *and let* $\epsilon \in (0, \frac{1}{10})$. *Then a uniform sample of size* $O(\epsilon^{-2} \log(d/\epsilon))$ *from* $P$ *is a stable* $(\epsilon, O(\epsilon))$-*coreset for* 1-*median in Jaccard metric with probability at least* $\frac{4}{5}$.

This result provides the first coreset construction based on uniform sampling for the Jaccard metric. Prior work implies a strong coreset of size $\tilde{O}(\epsilon^{-4} k^2)$, because it holds for $k$-median in $\ell_1$ by Jiang et al. (2024), however its construction algorithm must read the entire dataset.

**Approximation algorithms for $k$-median.** In metric spaces that admit stable coresets through uniform sampling, we can employ the framework introduced by Kumar et al. (2004) and further refined for additional metric spaces by Ackermann et al. (2010). These papers design approximation algorithms for $k$-median in metric spaces that have the property that for every instance $P$, a 1-median of $P$ is approximated (with high probability) by an optimal, or approximately optimal, solution for a uniform sample $Q \subseteq P$. We restate the main theorem in (Ackermann et al., 2010) using the language of stable coresets, and it can now be applied to several metrics where it was previously unknown, including Hamming, Kendall-tau and Jaccard, thereby extending existing 1-median algorithms to the more general $k$-median problem.

**Theorem 5.4** (Theorem 1.1 in (Ackermann et al., 2010)). *Let $\epsilon \in (0, \frac{1}{5})$ and let $(\mathcal{X}, \mathrm{dist})$ be a metric space such that*

- *$\mathcal{X}$ admits a stable $(\epsilon, O(\epsilon))$-coreset through uniform sampling of size $s_\epsilon$ with constant probability; and*

- *1-median in $\mathcal{X}$ admits a $(1 + \epsilon)$-approximation algorithm that runs on input of size $n'$ in time $f(n', \epsilon)$.*

*Then $k$-median in $\mathcal{X}$ admits a $(1 + O(\epsilon))$-approximation algorithm that runs on input of size $n$ in time $f(s_\epsilon, \epsilon) \cdot (ks_\epsilon/\epsilon)^{O(ks_\epsilon)} n$.*

An alternative approach for $\ell_1$ metrics, proposed in (Jiang et al., 2024), is to build a strong coreset of size $\tilde{O}(\epsilon^{-4}k^2)$ and then solve the coreset instance by enumerating over all its $k$-partitions. In both approaches the running time is exponential in the coreset size, and in many reasonable settings, this exponential term is larger than the input size $n$ and thus dominates the total running time.

**$C$-dispersed instances.** We can actually refine Theorem 1.4 to prove that when the input $P$ satisfies a certain technical condition, uniform sampling yields a strong coreset, rather than merely a stable coreset. This refinement is particularly valuable when $(1 + \beta)$-approximation algorithms, for $\beta \gg \epsilon$, are available, because applying such an algorithm on the strong coreset achieves $(1 + \beta)(1 + O(\epsilon))$-approximation for the original input $P$, offering a significant speedup with marginal increase in error, compared to applying that same algorithm directly on $P$. This advantage becomes especially significant for in discrete metric spaces—such as Kendall-tau and Jaccard—where the median problem is NP-hard. Although PTAS algorithms exist for these metrics, they are often complex to implement and computationally expensive, making them impractical. In such scenarios, constant-factor approximation algorithms and heuristics offer a more accessible and efficient alternative.

The technical condition we require is quite simple, and has been used in some literature without defining or stating it explicitly. We say that an instance $P$ is $C$-*dispersed* for $C \geq 1$ if its diameter is at most $C$ times the average distance inside it, that is,

$$\max_{x,y \in P} \|x - y\|_1 \leq C \cdot \frac{1}{|P|^2} \sum_{x,y \in P} \|x - y\|_1 .$$

**Theorem 5.5.** *Let $P \subset \mathbb{R}^d$ be finite and $C$-dispersed, and let $\epsilon \in (0, \frac{1}{5})$. Then a uniform sample of size $O(C\epsilon^{-2} \log \frac{d}{\delta})$ from $P$ is a strong $\epsilon$-coreset for 1-median in $\ell_1^d$ with probability at least $1 - \delta$.*

## 6 EXPERIMENTS

We demonstrate the empirical effectiveness of stable coresets for the median problem through experiments on real-world datasets across different metrics, comparing their performance against importance-sampling methods. Our evaluation measures the relative error between the cost of a solution computed on the coreset and the cost of a solution computed by the same method on the original dataset, that is,

$$\widehat{E} = \frac{\mathrm{cost}(\hat{c}^Q, P) - \mathrm{cost}(\hat{c}^P, P)}{\mathrm{cost}(\hat{c}^P, P)}, \tag{6}$$

where $\hat{c}^Q$ is a center computed for the coreset $Q$, and $\hat{c}^P$ is a center computed for the original dataset $P$. This relative error is expressed as a percentage. In our experiments, we examine points in $\mathbb{R}^d$ endowed with the $\ell_1$ metric as well as permutations under the Kendall-tau metric. For points in $\mathbb{R}^d$, we efficiently compute the optimal median by taking the coordinate-wise median, while for permutations we employ either heuristic methods or an Integer Linear Programming (ILP) approach.

**Experimental setup.** All experiments were conducted on a PC with Apple M1 and 16GB RAM running Python 3.9.6 on Darwin 22.6.0. For each experiment, we report the average results over 20 independent runs to ensure statistical significance. The datasets used in our experiments are detailed in Table 1. The source code used to run the experiments is available at https://github.com/amircarmel-lab/StableCoresets.

Table 1: Specifications of datasets used in Section 6.

| Dataset | Size $n$ | Dim. $d$ | Description |
|---|---|---|---|
| Yellow Taxi NYC (YT) (NYC Taxi and Limousine Commission, 2024) | 2.8M | 11 | New York City taxi trips in Jan. 2024 |
| Twitter (Chan et al., 2018) | 1.3M | 3 | Timestamp, latitude, longitude of tweets |
| Single-Cell Gene Expression (SCGE) (10x Genomics, 2019) | 7,865 | 33,586 | Peripheral blood mononuclear cells gene expression |
| My Anime List (MAL) (Valdivieso, 2020) | 234K | 50 | User rankings of anime titles |

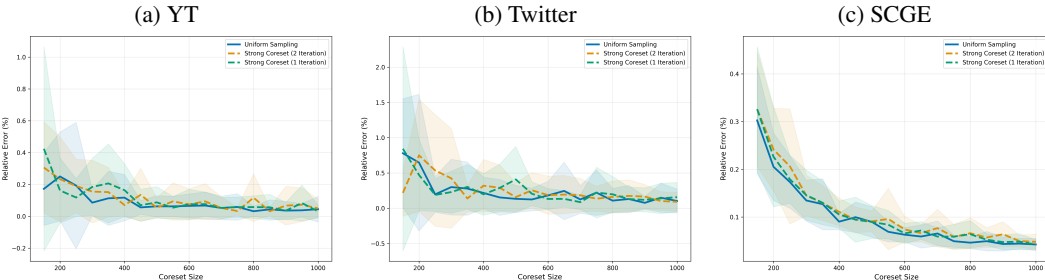

Figure 1: Tradeoff between coreset size and relative error, comparing importance sampling-based coresets with uniform sampling-based coresets across three datasets. Shaded regions represent one standard deviation.

**Experiment 1: Comparison with importance sampling.** We compare uniform sampling against the importance sampling-based coreset construction proposed by Jiang et al. (2024). Their method works by iteratively computing sensitivity scores for each point and sampling progressively smaller subsets, with each iteration reducing the size by a logarithmic factor until achieving a dimension-independent coreset. In our implementation, we evaluated their approach using both one and two iterations of this reduction process. The comparison focuses on the 1-median problem in $\ell_1$ metric across three datasets from Table 1; Yellow Taxi NYC, Twitter and Single-Cell Gene Expression.

Figure 1 demonstrates that uniform sampling achieves comparable error rates to importance sampling across all datasets. This efficiency difference is fundamental: importance sampling requires examining the entire dataset, resulting in construction time linear in the dataset size, while uniform sampling requires only constant time per sample and is completely dataset-oblivious—it requires no inspection or processing of the dataset prior to sampling. In our experiments, importance sampling took approximately $82/114/512$ seconds for datasets YT/Twitter/SCGE respectively, whereas uniform sampling required only $0.0001$ seconds to sample 500 points. The shaded regions represent one standard deviation, demonstrating comparable variability between the two methods.

**Experiment 2: Heuristic for Kendall-tau distance.** Using the MAL dataset, we aggregated rankings of $234K$ users over $50$ anime titles. We implemented five widely used rank aggregation methods: three Markov Chain-based approaches (MC1, MC2, MC3), Borda's sorting algorithm and scaled footrule aggregation (SFO) (Dwork et al., 2001; Kaur et al., 2017). Figure 2a illustrates the relative error of solutions computed on coresets of different sizes. It is important to note that these heuristics do not directly optimize the Kendall-tau cost. Consequently, solutions computed on the coresets can occasionally yield lower Kendall-tau costs than those from the original dataset, resulting in negative relative error values. The results confirm that relatively small coresets achieve results comparable to those obtained on the original dataset, even when using heuristic approaches.

**Experiment 3: Fairness constraints.** In this experiment, we sampled coresets of varying sizes from the dataset and then applied fairness constraints based on an arbitrary partitioning of items into two groups. We implemented the fairness-constrained integer linear programming algorithm by Kuhlman & Rundensteiner (2020) on both the original dataset and the coresets. Due to algorithmic constraints, we restricted the dataset to 8,700 user rankings, with each user ranking 16 anime titles. This algorithm optimizes the Kendall-tau cost objective while enforcing the fairness measure as

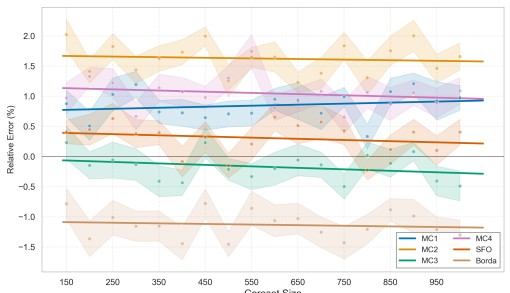
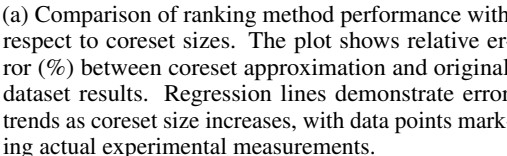
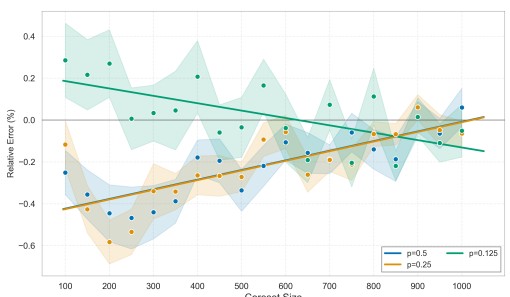

(a) Comparison of ranking method performance with respect to coreset sizes. The plot shows relative error (%) between coreset approximation and original dataset results. Regression lines demonstrate error trends as coreset size increases, with data points marking actual experimental measurements.

(b) Impact of fairness constraints on coreset approximation error. The parameter $p$ represents the probability of sampling popular anime titles across two distinct groups. $p = 0.5$ indicates balanced sampling between groups, while lower $p$ values indicate one group containing predominantly less popular anime. Shaded regions represent the standard error of the mean across multiple experimental runs.

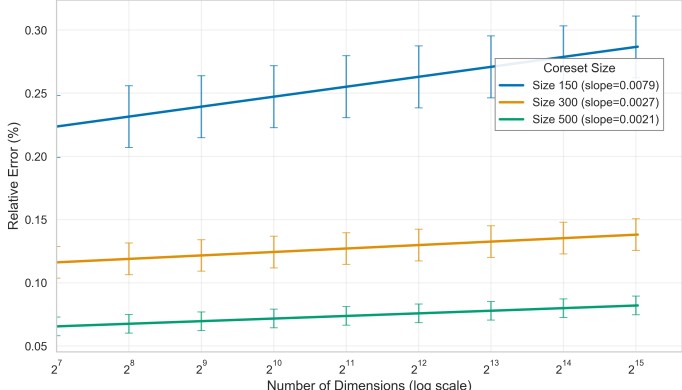

(c) Relative error (%) versus number of dimensions (log scale) for different coreset sizes. Regression lines demonstrate error trends as the number of dimensions increases. The slope is outlined for each regression line. Note that as the coreset size increases, the variance decreases, making the slope coefficient more statistically meaningful.

a linear constraint. Negative error values occur because the ILP solver might find slightly better solutions on the coreset rather than on the original dataset. Figure 2b demonstrates that solutions obtained from even small-sized coresets closely approximate those from the original dataset. This confirms that uniform sampling produces stable coresets that effectively support constraint-based optimization, even when those constraints were not considered during the sampling process.

**Experiment 4: Dimension dependency.** While our theoretical analysis establishes stable coreset size bounds that depend on the dimension $d$ (Theorem 1.4), we conjecture that this dependency is unnecessary. This experiment specifically tests whether the uniform sampling coreset performance remains dimension-independent.

To test this, we utilized the high-dimensional Single-Cell Gene Expression dataset (7,865 samples across 33,586 dimensions). For a given dimension count $d$, we randomly selected $d$ dimensions from the input dataset and then uniformly sampled a coreset for various sizes (150, 300, 500). For each coreset, we measured the relative error as defined in Equation 6.

Figure 2c illustrates the relationship between dimension count and error rates. We note that the slight upward trend in error is likely attributable to sampling variance rather than dimensional dependency. This demonstrates that the relative error remains stable as dimension count increases, supporting our conjecture that the theoretical bounds could be tightened to yield a dimension-independent bound.

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

## A    PROOFS OMITTED FROM SECTION 2

**Proposition 2.1.** *Let $(\mathcal{X}, \mathrm{dist})$ be a metric space and let $P \subseteq \mathcal{X}$ be a 1-median instance.*

*(a). Every stable $(\epsilon, \eta)$-coreset of $P$ is also a weak $(\epsilon, \eta)$-coreset.*

*(b). Every strong $\epsilon$-coreset of $P$, for $\epsilon \in (0, \frac{1}{5})$, is also a stable $(\epsilon, 4\epsilon)$-coreset.*

*Proof.* To prove Item (a), let $c^P, c^Q$ denote optimal medians for $P, Q$ in $\mathcal{X}$, respectively. Consider $c \in \mathcal{X}$ with $\mathrm{cost}(c, Q) \leq (1+\epsilon) \mathrm{opt}(Q)$. Then, $\mathrm{cost}(c, Q) \leq (1+\epsilon) \mathrm{cost}(c^Q, Q) \leq (1+\epsilon) \mathrm{cost}(c^P, Q)$. Then, following Equation 3, $\mathrm{cost}(c, P) \leq (1 + \eta) \mathrm{cost}(c^P, P)$.

To prove Item (b), let $c_1, c_2 \in \mathcal{X}$ such that $\mathrm{cost}(c_1, Q) \leq (1+\epsilon) \mathrm{cost}(c_2, Q)$. Following the definition of strong $\epsilon$-coreset, it follows that:

$$
\begin{aligned}
\mathrm{cost}(c_1, P) &\leq \frac{\mathrm{cost}(c_1, Q)}{1 - \epsilon} \leq \frac{(1 + \epsilon) \mathrm{cost}(c_2, Q)}{1 - \epsilon} \\
&\leq \frac{(1 + \epsilon)(1 + \epsilon)}{1 - \epsilon} \mathrm{cost}(c_2, P) \leq (1 + 4\epsilon) \mathrm{cost}(c_2, P).
\end{aligned}
$$

$\square$

**Fact 2.2.** *Let $f : \mathcal{X}_1 \rightarrow \mathcal{X}_2$ be an isometric embedding between metric spaces $(\mathcal{X}_1, \mathrm{dist}_1)$ and $(\mathcal{X}_2, \mathrm{dist}_2)$. Then,*

*(a). $f$ is injective; and*

*(b). for every $P \subseteq \mathcal{X}_1$ and $c \in \mathcal{X}_1$, $\mathrm{cost}(c, P) = \mathrm{cost}(f(c), f(P))$.*

*Proof.* For Item (a), if $x_1 \neq x_2$, then $\mathrm{dist}_1(x_1, x_2) \neq 0$, thus $f(x_1) \neq f(x_2)$. For Item (b), $\mathrm{cost}(c, P) = \sum_{p \in P} \mathrm{dist}_1(c, p) = \sum_{p \in P} \mathrm{dist}_2(f(c), f(p)) = \mathrm{cost}(f(c), f(P))$. $\square$

**Proposition 2.3.** *Let $f : \mathcal{X}_1 \rightarrow \mathcal{X}_2$ be an isometric embedding between metric spaces $(\mathcal{X}_1, \mathrm{dist}_1)$ and $(\mathcal{X}_2, \mathrm{dist}_2)$. For every $Q \subseteq P \subseteq \mathcal{X}_1$, if $f(Q)$ is a stable $(\epsilon, \eta)$-coreset of $f(P)$ in $\mathcal{X}_2$, then $Q$ is a stable $(\epsilon, \eta)$-coreset of $P$ in $\mathcal{X}_1$.*

*Proof.* Let $c_1, c_2 \in \mathcal{X}_1$ such that $\mathrm{cost}(c_1, Q) \leq (1 + \epsilon) \mathrm{cost}(c_2, Q)$. By Fact 2.2, this implies $\mathrm{cost}(f(c_1), f(Q)) \leq (1 + \epsilon) \mathrm{cost}(f(c_2), f(Q))$. Consequently, we have $\mathrm{cost}(f(c_1), f(P)) \leq (1 + \eta) \mathrm{cost}(f(c_2), f(P))$. Applying Fact 2.2 again yields $\mathrm{cost}(c_1, P) \leq (1 + \eta) \mathrm{cost}(c_2, P)$. $\square$

### A.1    STABLE CORESETS IN FINITE METRIC SPACES

We remind the reader the following result by Indyk (2001).

**Theorem A.1** (Theorem 31 in (Indyk, 2001))**.** *Let $\epsilon \in (0, 1)$, and $Q$ be a random sample from $P$. For arbitrary pair of points $a, b \in \mathcal{X}$, if $\mathrm{cost}(a, P) > (1 + \epsilon) \mathrm{cost}(b, P)$, then*

$$
\Pr[\mathrm{cost}(a, Q) > \mathrm{cost}(b, Q)] \geq 1 - e^{-\epsilon^2 |Q|/64}.
$$

A folklore analysis based on Thoerem A.1 leads to the following result.

**Corollary A.2.** *Fix a finite metric space $\mathcal{X}$ and let $P \subset \mathcal{X}$. Then, for every $\epsilon, \delta \in (0, 1)$, a uniform sample $Q \subseteq P$ of size $|Q| \geq 64\epsilon^{-2}(2 \ln |\mathcal{X}| + \ln(1/\delta))$ is a stable $(0, \epsilon)$-coreset with probability at least $1 - \delta$.*

*Proof.* Let $S$ contain all unordered pairs $a, b \in \mathcal{X}$ such that $\mathrm{cost}(a, P) > (1 + \epsilon) \mathrm{cost}(b, P)$. For each such pair, let $A_{a,b}$ be the event that $\mathrm{cost}(a, Q) \leq \mathrm{cost}(b, Q)$, Then by applying Theorem A.1 and a union bound,

$$
\Pr[\,Q \text{ is not a stable } (\epsilon, 0)\text{-coreset}\,] \leq \Pr[\cup_{a,b \in S} A_{a,b}] \leq \sum_{a,b \in S} \Pr[A_{a,b}] \leq |\mathcal{X}|^2 e^{-\epsilon^2 \frac{|Q|}{64}} \leq \delta.
$$

$\square$

## B   PROOFS OMITTED FROM SECTION 3

**Theorem 3.1.** *Let $P \subseteq \mathcal{X}$ and $\epsilon \in (0, \frac{1}{5})$, and suppose $\overline{\mathrm{cost}}(\mu, Q) \leq c \cdot \overline{\mathrm{cost}}(\mu, P)$ for some $c \geq 1$. If $Q \subseteq P$ is an $\epsilon$-RCDA of $P$ in $\mathcal{X}$ then $Q$ is a $(\frac{\epsilon}{c}, 4\epsilon)$-stable coreset of $P$.*

*Proof.* Let $x, y \in \mathcal{X}$ such that $\overline{\mathrm{cost}}(x, P) > (1 + 4\epsilon)\,\overline{\mathrm{cost}}(y, P)$. Using (4), we have

- $\overline{\mathrm{cost}}(x, Q) \geq (1 - \epsilon)\,\overline{\mathrm{cost}}(x, P) - \overline{\mathrm{cost}}(\mu, P) + \overline{\mathrm{cost}}(\mu, Q)$.
- $\overline{\mathrm{cost}}(y, Q) \leq (1 + \epsilon)\,\overline{\mathrm{cost}}(y, P) - \overline{\mathrm{cost}}(\mu, P) + \overline{\mathrm{cost}}(\mu, Q)$.

We can then derive

$$
\begin{aligned}
\overline{\mathrm{cost}}(x, Q) &\geq (1 - \epsilon)\,\overline{\mathrm{cost}}(x, P) - \overline{\mathrm{cost}}(\mu, P) + \overline{\mathrm{cost}}(\mu, Q) \\
&> (1 - \epsilon)(1 + 4\epsilon)\,\overline{\mathrm{cost}}(y, P) - \overline{\mathrm{cost}}(\mu, P) + \overline{\mathrm{cost}}(\mu, Q) \\
&= (1 + \epsilon + 2\epsilon(1 - 2\epsilon))\,\overline{\mathrm{cost}}(y, P) - \overline{\mathrm{cost}}(\mu, P) + \overline{\mathrm{cost}}(\mu, Q) \\
&\geq \overline{\mathrm{cost}}(y, Q) + 2\epsilon(1 - 2\epsilon)\,\overline{\mathrm{cost}}(y, P).
\end{aligned} \tag{7}
$$

Using (4) again, we can write

$$
(1 + \epsilon)\,\overline{\mathrm{cost}}(y, P) \geq \overline{\mathrm{cost}}(y, Q) + \overline{\mathrm{cost}}(\mu, P) - \overline{\mathrm{cost}}(\mu, Q) \geq \frac{1}{c}\,\mathrm{cost}(y, Q),
$$

where the last inequality follows from the fact that $\overline{\mathrm{cost}}(y, Q) \geq \overline{\mathrm{cost}}(\mu, Q)$ and $\overline{\mathrm{cost}}(\mu, Q) \leq c\,\overline{\mathrm{cost}}(\mu, P)$. Consequently,

$$
\overline{\mathrm{cost}}(y, P) \geq \frac{1}{c(1 + \epsilon)}\,\overline{\mathrm{cost}}(y, Q).
$$

Plugging this into (7) and using our assumption that $\epsilon < \frac{1}{5}$, we conclude that

$$
\overline{\mathrm{cost}}(x, Q) \geq \left(1 + \frac{2\epsilon(1 - 2\epsilon)}{c(1 + \epsilon)}\right)\overline{\mathrm{cost}}(y, Q) \geq (1 + \frac{\epsilon}{c})\,\overline{\mathrm{cost}}(y, Q).
$$

$\square$

## C   PROOFS OMITTED FROM SECTION 4

**Proposition 4.2.** $\lfloor \log d \rfloor \leq \mathrm{VCdim}(\mathcal{T}) \leq 2 \log d$.

*Proof.* For the upper bound, let $\mathrm{VCdim}(\mathcal{T}) = m$, then there are $m$ points $x_0, \ldots, x_{m-1} \in \mathbb{R}^d$ such that $V = \{(\tau(x_0), \ldots, \tau(x_{m-1})) : \tau \in \mathcal{T}\}$ is of size $2^m$. We restrict attention to tuples $(\tau(x_0), \ldots, \tau(x_{m-1})) \in V$ whose coordinates sum to $\lfloor \frac{m}{2} \rfloor$, denoted $V_{m/2} := \{v \in V : \sum_{i=0}^{m-1} v[i] = \lfloor \frac{m}{2} \rfloor\}$. For each $v \in V_{m/2}$, we select a threshold function $\tau \in \mathcal{T}$ that realizes this vector, i.e., such that $v = (\tau(x_0), \ldots, \tau(x_{m-1}))$. We denote this function by $\tau_{i_v, r_v}$, for some $i_v \in [d]$ and $r_v \in \mathbb{R}$. We claim that if $v_1 \neq v_2 \in V_{m/2}$ then $i_{v_1} \neq i_{v_2}$. To see this, assume without loss of generality that $r_{v_1} \leq r_{v_2}$. If $i_{v_1} = i_{v_2}$, then for every $x \in \mathbb{R}$, $\tau_{i_{v_2}, r_{v_2}}(x) = 1$ implies $\tau_{i_{v_1}, r_{v_1}}(x) = 1$. Since the coordinate sums of $v_1$ and $v_2$ are equal, we must have $v_1 = v_2$. Therefore, by the pigeonhole principle, $|V_{m/2}| \leq d$. However, using the known bound $|V_{m/2}| = \binom{m}{\lfloor \frac{m}{2} \rfloor} \geq \frac{1}{\sqrt{2m}} 2^m$, we get that $m \leq 2 \log d$.

For the lower bound, we construct a shattering set of points $x_0, \ldots, x_{m-1} \in \mathbb{R}^d$ where $m = \log d$ (assuming for simplicity that $d$ is a power of 2). We identify each coordinate with a binary vector $v \in \{0, 1\}^m$, corresponding to its binary representation, and define $x_i[v] = 1 - v[i]$. To show that $x_0, \ldots, x_{m-1}$ is a shattering set, consider the functions $(\tau_{v, \frac{1}{2}})_{v \in \{0,1\}^m} \subseteq \mathcal{T}$. For every $v \in \{0, 1\}^m$, we have:

$$
(\tau_{v, \frac{1}{2}}(x_0), \ldots, \tau_{v, \frac{1}{2}}(x_{m-1})) = (\mathbb{1}_{x_0[v] \leq \frac{1}{2}}, \ldots, \mathbb{1}_{x_{m-1}[v] \leq \frac{1}{2}}) = (\mathbb{1}_{v[0] \geq \frac{1}{2}}, \ldots, \mathbb{1}_{v[m-1] \geq \frac{1}{2}}) = v
$$

$\square$

Let $P \subset \mathbb{R}^d$ of size $n = |P|$, and assume for simplicity that $n$ is odd. Without loss of generality, we may assume that $P$ contains no repeated points, as duplicate points can be perturbed by an infinitesimal amount. Denote by $\mu \in \mathbb{R}^d$ a median of $P$. We will also need the following well-known fact.

**Fact C.1.** *Let $P \subset \mathbb{R}^d$ be finite. Then, $\mu \in \mathbb{R}^d$ is a 1-median for $P$ if and only if for every $i \in [d]$, $\mu[i]$ is a median value of the $i$-th coordinate of all points in $P$.*

**Lemma 4.6.** *Let $\epsilon \in (0, 1)$ and let $P$ be a finite set in $\ell_1^d$. If $Q$ is an $\epsilon$-approximation of $P$, then $Q$ is a $20\epsilon$-RCDA of $P$.*

*Proof.* We first consider $d = 1$, and without loss of generality assume $x \leq \mu$. To evaluate the difference in average costs, we break the calculation into the three regions $L$, $M$, and $U$ (similar strategy was used in (Danos, 2021)): $L := \{p \in P : p \leq x\}$, $M := \{p \in P : x < p \leq \mu\}$, and $U := \{p \in P : \mu < p\}$. In each region, we can simplify the expression $|x - p| - |\mu - p|$ by expressing the absolute values explicitly, which allows us to evaluate $\overline{\text{cost}}(x, P) - \overline{\text{cost}}(\mu, P)$. Notably, in regions $L$ and $U$, the value of $|x - p| - |\mu - p|$ depends only on $x$ and $\mu$, not on the specific value of $p$. Observe that $|L| = |P| \, \text{edf}_P(x)$, $|M| = |P| \, (\text{edf}_P(\mu) - \text{edf}_P(x))$ and $|U| = |P| \, (1 - \text{edf}_P(\mu)) = \frac{|P|}{2}$. We can write

$$
\overline{\text{cost}}(x, P) - \overline{\text{cost}}(\mu, P) = \frac{1}{|P|} \Big( \sum_{p \in L} |x - p| + \sum_{p \in M} |x - p| + \sum_{p \in U} |x - p| \Big) - \overline{\text{cost}}(\mu, P)
$$

$$
= \frac{1}{|P|} \Big( \sum_{p \in L} \big( |\mu - p| - |x - \mu| \big) + \sum_{p \in M} \big( |x - \mu| - |\mu - p| \big)
$$

$$
+ \sum_{p \in U} \big( |\mu - p| + |x - \mu| \big) \Big) - \overline{\text{cost}}(\mu, P)
$$

$$
= \frac{1}{|P|} \Big( - \sum_{p \in L} |x - \mu| + \sum_{p \in M} |x - \mu| + \sum_{p \in U} |x - \mu|
$$

$$
- 2 \sum_{p \in M} |\mu - p| \Big)
$$

$$
= |x - \mu|(1 - 2 \, \text{edf}_P(x)) - \frac{2}{|P|} \sum_{p \in M} |\mu - p|.
$$

Similarly for $Q$ (the fact that $\mu$ is a median of $P$ is not utilized in the argument above),

$$
\overline{\text{cost}}(x, Q) - \overline{\text{cost}}(\mu, Q) = |x - \mu|(1 - 2 \, \text{edf}_Q(x)) - \frac{2}{|Q|} \sum_{q \in M \cap Q} |\mu - q|,
$$

and thus

$$
\big( \overline{\text{cost}}(x, P) - \overline{\text{cost}}(\mu, P) \big) - \big( \overline{\text{cost}}(x, Q) - \overline{\text{cost}}(\mu, Q) \big)
$$

$$
= 2|x - \mu| \big( \text{edf}_Q(x) - \text{edf}_P(x) \big) - \frac{2}{|P|} \sum_{p \in M} |\mu - p| + \frac{2}{|Q|} \sum_{q \in M \cap Q} |\mu - q|. \tag{8}
$$

We will now bound each on of the terms in (8). The first term is bounded utilizing the $\epsilon$-approximation property of $Q$,

$$
-2\epsilon|\mu - x| \leq 2|x - \mu| \big( \text{edf}_Q(x) - \text{edf}_P(x) \big) \leq 2\epsilon|\mu - x|. \tag{9}
$$

We now bound the term $\frac{1}{|Q|} \sum_{q \in M \cap Q} |\mu - q| - \frac{1}{|P|} \sum_{p \in M} |\mu - p|$. Partition $M$ to intervals $I_0, I_1, ..., I_t$, such that each interval contains exactly $2\epsilon|P|$ points, except for the last one which might be smaller. We denote $I_i = (p_i, p_{i+1}]$, with $p_0 = x$ and $p_{t+1} = \mu$ and $a_i = \frac{1}{|Q|}|I_i \cap Q| - \frac{1}{|P|}|I_i \cap P|$ for $i = 0, ..., t$.

Here the idea is to partition $M$ into intervals containing a controlled number of points, leveraging the fact that $Q$ approximates the proportion of points in each interval. By bounding each interval's contribution using the interval endpoints and the relative difference in point counts between $P$ and $Q$,

then applying telescoping sums across all intervals, we can establish tight bounds on how much the average distances in $Q$ can deviate from those in $P$ across the entire region $M$.

Each interval contains at least 0 points from $Q$ and at most $4\epsilon|Q|$ points from $Q$. Consequently, $-2\epsilon \leq \sum_{i=j_1}^{j_2} a_i \leq 2\epsilon$, for every $j_1 \leq j_2$. We can write

$$\frac{1}{|Q|}\sum_{q\in M\cap Q}|\mu-q| - \frac{1}{|P|}\sum_{p\in M}|\mu-p| = \frac{1}{|Q|}\sum_{i=0}^{t}\sum_{q\in I_i\cap Q}|\mu-q| - \frac{1}{|P|}\sum_{i=0}^{t}\sum_{p\in I_i\cap P}|\mu-p|.$$

For the upper bound,

$$\frac{1}{|Q|}\sum_{i=0}^{t}\sum_{q\in I_i\cap Q}|\mu-q| - \frac{1}{|P|}\sum_{i=0}^{t}\sum_{p\in I_i\cap P}|\mu-p|$$

$$\leq \frac{1}{|Q|}\sum_{i=0}^{t}\sum_{q\in I_i\cap Q}|\mu-p_i| - \frac{1}{|P|}\sum_{i=0}^{t}\sum_{q\in I_i\cap P}|\mu-p_{i+1}|$$

$$\leq \frac{1}{|Q|}\sum_{i=0}^{t}|I_i\cap Q||\mu-p_i| - \frac{1}{|P|}\sum_{i=0}^{t}|I_i\cap P||\mu-p_{i+1}|$$

$$\leq \sum_{i=0}^{t}\left(a_i + \frac{1}{|P|}|I_i\cap P|\right)|\mu-p_i| - \frac{1}{|P|}\sum_{i=0}^{t}|I_i\cap P||\mu-p_{i+1}|$$

$$\leq \sum_{i=0}^{t}a_i|\mu-p_i| + \sum_{i=0}^{t}\frac{|I_i\cap P|}{|P|}(|\mu-p_i|-|\mu-p_{i+1}|)$$

$$\leq \sum_{i=0}^{t}a_i|\mu-p_i| + 2\epsilon\sum_{i=0}^{t}(p_{i+1}-p_i) \leq 4\epsilon|\mu-x|,$$

Similarly for the lower bound,

$$\frac{1}{|Q|}\sum_{i=0}^{t}\sum_{q\in I_i\cap Q}|\mu-q| - \frac{1}{|P|}\sum_{i=0}^{t}\sum_{q\in I_i\cap P}|\mu-p|$$

$$\geq \frac{1}{|Q|}\sum_{i=0}^{t}\sum_{q\in I_i\cap Q}|\mu-p_{i+1}| - \frac{1}{|P|}\sum_{i=0}^{t}\sum_{q\in I_i\cap P}|\mu-p_i|$$

$$\geq \sum_{i=0}^{t}a_i|\mu-p_{i+1}| + 2\epsilon\sum_{i=0}^{t}(p_i-p_{i+1})$$

$$\geq -4\epsilon|\mu-x|.$$

We get that

$$-8\epsilon|\mu-x| \leq -\frac{2}{|P|}\sum_{p\in M}|\mu-p| + \frac{2}{|Q|}\sum_{q\in M\cap Q}|\mu-q| \leq 8\epsilon|\mu-x|. \tag{10}$$

Thus, by combining both Equation 9 and 10, we obtain

$$-10\epsilon|\mu-x| \leq (\overline{\text{cost}}(x,P)-\overline{\text{cost}}(\mu,P)) - (\overline{\text{cost}}(x,Q)-\overline{\text{cost}}(\mu,Q)) \leq 10\epsilon|\mu-x|.$$

The case $d=1$ follows because $\overline{\text{cost}}(x,P) \geq \frac{1}{|P|}\sum_{p\in U}|x-p| \geq \frac{1}{|P|}\sum_{p\in U}|x-\mu| = \frac{|U|}{|P|}|x-\mu| = \frac{1}{2}|x-\mu|$, where the last equality uses $|U| = \frac{1}{2}|P|$.

The general case $d \geq 1$ follows immediately because $\overline{\text{cost}}(x,P) = \sum_{i=1}^{d}\overline{\text{cost}}_i(x,P)$, where $\overline{\text{cost}}_i(x,P) := \frac{1}{|P|}\sum_{p\in P}|x[i]-p[i]|$. This completes the proof of Lemma 4.6. $\qquad\square$

# D    PROOFS OMITTED FROM SECTION 5

**Stable coresets in Kendall-tau metric.**    To illustrate one concrete example of Theorem 1.4 in $\ell_1$ related metric space, consider the Kendall-tau metric on permutations. The Kemeny embedding maps each permutation $\sigma \in \mathcal{S}_d$ to a binary vector in $\{0,1\}^{\binom{d}{2}}$ where $\phi(\sigma)[i,j] = \mathbb{1}_{\sigma[i]<\sigma[j]}$. Applying Proposition 2.3 with Theorem 1.4 yields:

**Corollary D.1.** *Let $P \subset \mathcal{S}_d$ be finite and let $\epsilon \in (0, \frac{1}{5})$. Then a uniform sample of size $O(\epsilon^{-2}\log d)$ from $P$ is a stable $(\epsilon/6, 4\epsilon)$-coreset for 1-median in $\mathcal{S}_d$ under the Kendall-tau metric with probability at least $4/5$.*

We emphasize that only the existence of an embedding into $\ell_1$ is necessary; the explicit form of this embedding need not be known to derive these coreset guarantees.

## D.1    LOW-DISTORTION EMBEDDINGS

**Proposition 5.1.** *Let $f : \mathcal{X}_1 \to \mathcal{X}_2$ be an embedding between metric spaces $(\mathcal{X}_1, dist_1)$ and $(\mathcal{X}_2, dist_2)$ with distortion $D$ and scaling factor $r$. For every $Q \subseteq P \subseteq \mathcal{X}_1$, if $f(Q)$ is a stable $(\epsilon, \eta)$-coreset of $f(P)$ in $\mathcal{X}_2$ for some $\epsilon, \eta > 0$, and the values $\epsilon' := (1+\epsilon)/D^2 - 1$ and $\eta' := D^2(1+\eta) - 1$ are positive, then $Q$ is a stable $(\epsilon', \eta')$-coreset of $P$ in $\mathcal{X}_1$.*

*Proof.* Let $x, y \in \mathcal{X}_1$ such that $\mathrm{cost}(x, Q) \leq (1+\epsilon')\mathrm{cost}(y, Q)$. Using the fact that $f$ has distortion $D^2$ we obtain

$$\mathrm{cost}(f(x), f(Q)) \leq Dr\,\mathrm{cost}(x, Q) \leq Dr(1+\epsilon')\,\mathrm{cost}(y, Q)$$
$$\leq D^2(1+\epsilon')\,\mathrm{cost}(f(y), f(Q)) \leq (1+\epsilon)\,\mathrm{cost}(f(y), f(Q)).$$

Where the last inequality follows by our choice of $\epsilon'$. Since $f(Q)$ is stable $(\epsilon, \eta)$-coreset it follows that $\mathrm{cost}(f(x), f(P)) \leq (1+\eta)\,\mathrm{cost}(f(y), f(P))$. Using the properties of $f$ again

$$\mathrm{cost}(x, P) \leq \frac{D}{r}\,\mathrm{cost}(f(x), f(P)) \leq \frac{D}{r}(1+\eta)\,\mathrm{cost}(f(y), f(P))$$
$$\leq D^2(1+\eta)\,\mathrm{cost}(y, P) \leq (1+\eta')\,\mathrm{cost}(y, P),$$

where the last inequality follows by our choice of $\eta'$. $\qquad\square$

A *terminal embedding* with distortion $D^2 \geq 1$ between metric spaces $(\mathcal{X}_1, \mathrm{dist}_1)$ and $(\mathcal{X}_2, \mathrm{dist}_2)$ with respect to $P$ is a map $f : \mathcal{X}_1 \to \mathcal{X}_2$, such that for some $r > 0$,

$$\forall p \in P, \forall x \in \mathcal{X}_1, \qquad \frac{1}{D} \cdot \mathrm{dist}_2(f(p), f(x)) \leq r \cdot \mathrm{dist}_1(p, x) \leq D \cdot \mathrm{dist}_2(f(p), f(x)). \quad (11)$$

Clearly this is a weaker guarantee. Note that the proof of Proposition 5.1 holds immediately for this definition as well. Moreover, it is worth noting that in many practical scenarios, the distortion in Equation 5 is one-sided in which case the error propagation would occur only once, improving the approximation guarantees in Proposition 5.1.

## D.2    STABLE CORESETS IN EUCLIDEAN METRIC

Gordon refined Dvoretzky's Theorem and showed that $\ell_2^d$ embeds with distortion $1 + \epsilon$ into $\ell_1^{O(\epsilon^{-2}d)}$ (Gordon, 1988; Schechtman, 2006). Thus, Corollary 5.2 implies the following.

**Corollary D.2.** *Let $P \subseteq \mathbb{R}^d$ be finite and $\epsilon \in (0, \frac{1}{10})$. Then a uniform sample of size $O(\epsilon^{-2}\log(d/\epsilon))$ from $P$ is a stable $(\epsilon, O(\epsilon))$-coreset for 1-median in $\ell_2^d$ with probability at least $\frac{4}{5}$.*

This result provides a different tradeoff than prior work (Huang et al., 2023a), which showed that a uniform sampling of size $\tilde{O}(\epsilon^{-3})$ yields a weak $(\epsilon, O(\epsilon))$-coreset.

### D.3 $C$-DISPERSED INSTANCES

**Theorem 5.5.** *Let $P \subset \mathbb{R}^d$ be finite and $C$-dispersed, and let $\epsilon \in (0, \frac{1}{5})$. Then a uniform sample of size $O(C\epsilon^{-2} \log \frac{d}{\delta})$ from $P$ is a strong $\epsilon$-coreset for $1$-median in $\ell_1^d$ with probability at least $1 - \delta$.*

*Proof.* As before, let $\mu$ denote a median of $P$. By the triangle inequality,

$$\max_{x,y \in P} \|x - y\|_1 \leq \frac{C}{|P|^2} \sum_{x,y \in P} \|x - \mu + \mu - y\|_1 \leq \frac{2C}{|P|} \sum_{x \in P} \|\mu - x\|_1 = 2C \overline{\mathrm{opt}}(P),$$

and thus for every $x \in P$,

$$\|\mu - x\|_1 = \frac{1}{|P|} \sum_{y \in P} \|\mu - y + y - x\|_1 \leq \frac{1}{|P|} \sum_{y \in P} (\|\mu - y\|_1 + \|y - x\|_1)$$

$$\leq \overline{\mathrm{opt}}(P) + 2C \overline{\mathrm{opt}}(P) \leq (2C + 1) \overline{\mathrm{opt}}(P).$$

Additionally, $\mathbb{E}[\overline{\mathrm{cost}}(\mu, Q)] = \overline{\mathrm{cost}}(\mu, P) = \overline{\mathrm{opt}}(P)$, and thus Chernoff's bound implies

$$\Pr \left[ \left| \overline{\mathrm{cost}}(\mu, Q) - \overline{\mathrm{cost}}(\mu, P) \right| \geq \frac{\epsilon}{2} \overline{\mathrm{cost}}(\mu, P) \right] \leq 2 \exp \left( -\frac{\epsilon^2 |Q|}{3(2C + 1)} \right) \leq \frac{\delta}{2}.$$

Following Corollary 4.5 and Lemma 4.6, with probability at least $1 - \frac{\delta}{2}$, the subset $Q$ is $\frac{\epsilon}{2}$-RCDA. By the union bound, with probability at least $1 - \delta$, we have that $Q$ is $\frac{\epsilon}{2}$-RCDA and $\left| \overline{\mathrm{cost}}(\mu, P) - \overline{\mathrm{cost}}(\mu, Q) \right| \leq \frac{\epsilon}{2} \overline{\mathrm{cost}}(\mu, P)$. In this case, for every $x \in \mathbb{R}^d$,

$$\left| \overline{\mathrm{cost}}(x, P) - \overline{\mathrm{cost}}(x, Q) \right|$$

$$\leq \left| [\overline{\mathrm{cost}}(x, P) - \overline{\mathrm{cost}}(\mu, P)] - [\overline{\mathrm{cost}}(x, Q) - \overline{\mathrm{cost}}(\mu, Q)] \right| + \left| \overline{\mathrm{cost}}(\mu, P) - \overline{\mathrm{cost}}(\mu, Q) \right|$$

$$\leq \frac{\epsilon}{2} \overline{\mathrm{cost}}(x, P) + \frac{\epsilon}{2} \overline{\mathrm{cost}}(\mu, P) \leq \epsilon \overline{\mathrm{cost}}(x, P),$$

which establishes that $Q$ is a strong $\epsilon$-coreset. $\qquad \square$

