# OpenReview forum: "Stable coresets: Unleashing the power of uniform sampling"
_ICLR.cc/2026/Conference — ICLR 2026 Poster_

### Official Review · Reviewer_DWK3 · 2025-10-20

**Soundness:** 2
**Presentation:** 1
**Contribution:** 2
**Rating:** 4
**Confidence:** 3

**Summary:**

The paper introduces a new notion of coresets for the 1-median problem, termed stable coresets, whose power lies between that of strong and weak coresets. They construct a stable coreset of size $O(\log d / \epsilon^2)$ for the 1-median problem in the $\ell_1$ metric. Using standard embedding results, they extend these constructions to other metrics such as the Jaccard, Kendall-Tau, and Euclidean metrics. By combining their coresets with existing algorithms, they obtain approximation schemes in the corresponding metric spaces whose running times are FPT in $k$, $\epsilon$ and the coreset size. Finally, they present experimental results evaluating their constructions and comparing them with existing ones.

The main construction is based on a simple procedure that produces a uniform sample of size $O(\log d / \epsilon^2)$. The proof appears to rely on several existing results combined in sequence. The key motivation for using uniform sampling in constructing stable coresets is its running time---uniform sampling can be performed in sublinear time, in contrast to existing approaches for strong coresets, which examine all points in the input.

**Strengths:**

1. The newly proposed notion of coresets: stable coresets, lies between weak and strong coresets in terms of power, which makes it appealing both theoretically and practically.
2. The construction is quite simple: a uniform sample of an appropriate size suffices to obtain a stable coreset.

**Weaknesses:**

1. The paper motivates the design of coresets in sublinear time and shows that uniform sampling yields a stable coreset for the 1-median problem in the $\ell_1$ metric. However, it is not clear what stable coresets are useful for beyond designing EPASes—where the sublinear-time motivation is largely lost. For example, why should stable coresets be preferred over weak coresets when aiming for sublinear running time? Furthermore, even if a stable coreset can be obtained in sublinear time, it is unclear how to evaluate the cost of centers on the original dataset in sublinear time.

2. A major weakness, in my view, is the technical writing. It is difficult to identify the key technical challenges and the insights that make the results work. This issue is becomes worse due to the fact that only one proof appears in the main paper, while all others—including one of the main theorems (Theorem 3.1)—are deferred to the appendix. While I understand the page constraints, it is important to at least sketch the proofs or highlight the main ideas behind the crucial technical statements. I also went through the proofs in the appendix, but they too fail to convey the underlying intuition. Moreover, the technical overview section does not highlight any novel contributions. Without this, it is difficult to assess the technical depth of the paper.

3. Related to point #2, the proof of Theorem 1.4 (one of the main results) seems to follow directly from existing results. It appears that this theorem is derived primarily by chaining together prior work. While this is not inherently bad, it gives the impression that the paper is mainly a collection of observations rather than a technically deep contribution, which weakens its case for a venue like ICLR.

4. Strong and weak coresets of size independent of the dimension are already known, whereas the coreset construction in this paper depends logarithmically on the dimension.

5. The proposed coreset result appears to apply only to the 1-median problem. However, most existing coreset results in specific metric spaces are designed to handle the more general $k$-median case.

6. As an application, the paper shows how to obtain approximation scheme for $k$-median in the $\ell_1$ metric using existing frameworks. However, the authors themselves suggest that such algorithms already exist, so it is unclear what new insight or contribution this application provides.

**Questions:**

1. How is designing stable coresets in sublinear time useful? For example, weak coresets can already be constructed in sublinear time. In that case, how does the additional power of stable coresets over weak coresets translate into any practical advantage?

2. In Theorem 5.5, you mention that it gives an approximation scheme for $k$-median in the $\ell_1$ metric with running time $f(s,\epsilon) \cdot (s k / \epsilon)^{O(ks)} \cdot n$, for a stable coreset of size $s$. How does this yield an EPAS (FPT in $k$ and $\epsilon$) for $k$-median in the $\ell_1$ metric? You mention that existing strong coresets can yield EPASes, but I do not see how the running time of your approach can be made independent of the dimension $d$.

3. You mention potential applications of stable coresets in fair clustering, but I do not see any technical results related to fair clustering in the paper.

4. Regarding Theorem 5.5: does this result come directly from prior literature? It refers to stable coresets, which is one of the contributions of this paper.

5. I am curious if it is possible to extend the notion of stable coresets to the $k$-median problem?

6. Concerning the proof of Proposition 2.1(a), I am unable to follow the argument, particularly the inequality in line 656. Could you please clarify this step?

---

> ### Author Response · Authors · 2025-11-19
>
> We sincerely thank all reviewers for their thoughtful feedback and constructive comments. Let us first address a common point regarding our focus on $k=1$, and then respond separately to each reviewer's specific questions.
>
> Our work focuses on uniform sampling as a means to construct coresets for the median problem, providing the first sublinear-time method for important discrete metrics such as Kendall-tau and Jaccard. To enable truly uniform sampling, we restrict attention to $k=1$ because it is well-known that for larger $k$, small clusters cannot be reliably sampled. Extensions to $k>1$ via uniform sampling are possible but are more limited and somewhat cumbersome. For example, Huang et al. (2023a) restrict the dataset (the optimal clustering must be balanced) and also modify the coreset definition (to consider only balanced clusterings of the coreset $Q$). We chose to focus on $k=1$ in order to construct a stable coreset, without any such additional modifications. We stress that results do have applications (even if not for coresets) for $k>1$. Indeed, Theorem 5.4 shows an application to linear-time approximation algorithms for small $k>1$; it uses the framework of Ackermann et al. (2010), which happens to require exactly what our coresets provide (that uniform sampling works well for $k=1$).
>
> Beyond these technical considerations, the $k=1$ case is itself an aggregation task that is fundamental. It asks for a single point that represents well the entire dataset, which is important and useful by itself. In particular, it arises in several important metric spaces and was given specialized names, e.g., rank aggregation (when points are permutations), the median string problem, the genome median problem, and Wasserstein barycenter (under earthmover aka optimal transport distance). In fact, some of our experiments deal with rank aggregation.
>
> We find vast potential in the stable coreset formulation and are encouraged by your positive review of it, and we believe other valuable results will follow from this new concept.
>
> **Response to questions:**
>
> 1. First, stable coresets naturally align with embeddability between metrics, allowing results in one metric space to be translated to another metric space. This allows us to obtain other coreset results that were previously unknown.
> Second, stable coresets permit imposing constraints over the set of possible solutions, such as in the fair rank aggregation problem, which seeks a consensus ranking satisfying certain fairness constraints (see the paragraph starting at line 057 and Experiment 3). This property does not hold for weak coresets.
> Third, the currently known results about weak coresets for the $\ell_1$ metric provide a guarantee only for an optimal solution over the coreset, while our result allows even for an approximate solution over the coreset. Danos was unable to extend his approach to this more general condition.
>
> 2. Currently, the bound for stable coresets does depend on the dimension, basically $O(\log d)$. The time bound in Theorem 5.5 is exponential in the coreset size, and thus in $O(\log d)$. Nevertheless, the dependence on $n$ is linear.
> Compared to previous approaches, our result has better dependence on $\epsilon$ and $k$ and will thus be better in some parameter regimes, e.g., when $\log d$ is small which is common in practical settings.
>
> 3. The term “fair clustering” might have multiple meanings in the literature, and we thus avoid using it (we will eliminate the one occurrence in the abstract). We clarify that by “fair” we mean that median must satisfy some “fairness constraint”, e.g., a ranking of people that is in itself gender-balanced.
>
> 4. You probably refer to Theorem 5.4, which indeed follows directly from Ackermann et al. (2010). While they do not use the term coreset, their framework basically requires that (for $k=1$) a near-optimal solution for a uniform sample is also near-optimal for the entire dataset. We show that uniform sampling indeed works in this way (our proof uses stable coresets, which is only stronger), and thus the framework of Ackermann et al. is immediately applicable. Our contribution is to show that this property holds in various metrics (such as Kendall-tau, Jaccard) where it was previously unknown.
>
> 5. Indeed, the definition can be immediately extended to $k$-median by replacing $c_1, c_2 \in \mathcal{X}$ with $C_1, C_2 \in \binom{\mathcal{X}}{k}$. We focused on the 1-median variant to obtain results via uniform sampling, which works well for $k=1$ (as discussed above).
>
> 6. The many letters might be confusing, but the two inequalities in line 656 are really simple. The first inequality follows from the assumption that $c$ is near-optimal for $Q$. The second inequality holds because $c^Q$ is optimal (for $Q$) and thus only better than $c^P$.

---

### Official Review · Reviewer_sxjp · 2025-10-27

**Soundness:** 3
**Presentation:** 4
**Contribution:** 3
**Rating:** 8
**Confidence:** 3

**Summary:**

In this paper, the authors propose an intermediate notion between strong coresets and weak coresets, which they call *stable coresets*. Roughly, stable coresets preserve the relative order of the costs across different centers: this is weaker than strong coresets, which preserve the cost of every center, and stronger than weak coresets, which only ensure that an optimal solution for the coreset is near-optimal for the original dataset, without necessarily preserving pairwise orders or cost values.

Importantly, stable coresets (1) can be easier to construct than strong coresets and (2) inherit a useful property of strong coresets: if a metric space admits a strong coreset, then every submetric also admits a strong coreset, i.e., coreset results for one metric space $\mathcal{X}$ extend to every metric that can be embedded isometrically in $\mathcal{X}$. This allows one to indirectly construct coresets for metrics that are embeddable into "easier" metrics, even if their direct coreset analysis is complicated.

The authors give a general recipe for proving stable coresets by showing a uniform bound on the cost difference for all centers (which they call an "$\epsilon$-relative cost-difference approximation (RCDA)".) They then instantiate this by showing that a uniform sample of size $O(\log d / \epsilon^2)$ yields a stable coreset for 1-median in $\mathbb{R}^d$ under the $\ell_1$ metric with constant probability. Since many metrics (e.g., Kendall-tau, Jaccard, tree metrics, and $\ell_2$) embed (nearly) isometrically into $\ell_1$​, they leverage the property of stable coresets to derive coreset constructions for those metrics. In addition, they show applications to fair clustering and to approximation algorithms for k-median in these metric spaces, and provide experimental results demonstrating the performance of stable coresets in practice (in terms of both construction time and approximation quality).

**Strengths:**

I like the formulation of stable coresets, especially how it naturally aligns with embeddability between metrics. (While a similar comparative guarantee for coresets appeared previously in [Indyk 1999, 2001], it was presented more as an algorithmic primitive rather than as a standalone definition. I therefore regard the stable-coreset formulation as novel.) The paper already demonstrates the usefulness of this formulation by constructing the first coresets based on uniform sampling for important metric spaces, including Kendall–tau and Jaccard, as well as new bounds for $\ell_2$. And one can imagine that such a "coreset-via-embedding" recipe will lead to further coreset results.


The paper is well written with a nice logical flow: (1) The motivation and goal are clear. (2) In the main body, the paper presents a general framework followed by a concrete instantiation. While the ideas are easy to follow, the proof of $\epsilon$-RCDA for $\ell_1$ via $\epsilon$-approximations is neat and non-trivial.

**Weaknesses:**

Although stable coresets provide a stronger guarantee than weak coresets (and, as mentioned above, I find the guarantee aesthetically interesting in its own right), at this point the main gain seems to be that given a near-isometric embedding, one can translate a coreset in the “host space” to a coreset in the “guest space.” From this perspective, a stable coreset does not offer additional advantage over a weak coreset *within the original host space* ($\ell_1$ in this paper), while the required coreset size given shown in this paper is larger compared to the known $\tilde{O}(1/\epsilon^2)$ size of weak coresets in $\ell_1$.

-----------------------------------------

Some editorial notes:

- Fact 2.2 (b) for every $c \in P$: I think you need for every $c \in \mathcal{X}_1$ to show the later Proposition 2.3?

- I'm not sure why it is necessary to include a proof to Proposition 4.2, given that the proposition already appears in (Gey, 2018)?

- In Theorem 4.4 it might be clearer to say explicitly that the random sample $Q \sim \mathcal{D}$.

- Equation 8 and 9: $edf_{p} \rightarrow edf_{P}$

- Line 811 "$-2\epsilon \leq a_i \leq 2\epsilon$": I suggest replacing this with just the summation bound "$-2\epsilon \leq \sum^{j_2}_{i=j_1}a_i \leq 2\epsilon$ for every $j_2 \geq j_1$" on Line 837, since the former is not used and is subsumed by the latter.

- Line 823 and onwards: Is $p_i$ ever defined? One can infer it is the boundary of the ith interval, but since it is the key object in the telescoping, I think it should be defined explicitly.

**Questions:**

Syntactically, the idea of partitioning the range of a coordinate into three different sets and intervals in the main technical proof (Lemma 4.6) appears a bit similar to the proof strategy in [Danos, 2021] (e.g., their proof of Lemma 2.2.6) Was this inspired by that work?

---

> ### Author Response · Authors · 2025-11-19
>
> We sincerely thank all reviewers for their thoughtful feedback and constructive comments. Let us first address a common point regarding our focus on $k=1$, and then respond separately to each reviewer's specific questions.
>
> Our work focuses on uniform sampling as a means to construct coresets for the median problem, providing the first sublinear-time method for important discrete metrics such as Kendall-tau and Jaccard. To enable truly uniform sampling, we restrict attention to $k=1$ because it is well-known that for larger $k$, small clusters cannot be reliably sampled. Extensions to $k>1$ via uniform sampling are possible but are more limited and somewhat cumbersome. For example, Huang et al. (2023a) restrict the dataset (the optimal clustering must be balanced) and also modify the coreset definition (to consider only balanced clusterings of the coreset $Q$). We chose to focus on $k=1$ in order to construct a stable coreset, without any such additional modifications. We stress that results do have applications (even if not for coresets) for $k>1$. Indeed, Theorem 5.4 shows an application to linear-time approximation algorithms for small $k>1$; it uses the framework of Ackermann et al. (2010), which happens to require exactly what our coresets provide (that uniform sampling works well for $k=1$).
>
> Beyond these technical considerations, the $k=1$ case is itself an aggregation task that is fundamental. It asks for a single point that represents well the entire dataset, which is important and useful by itself. In particular, it arises in several important metric spaces and was given specialized names, e.g., rank aggregation (when points are permutations), the median string problem, the genome median problem, and Wasserstein barycenter (under earthmover aka optimal transport distance). In fact, some of our experiments deal with rank aggregation.
>
> We find vast potential in the stable coreset formulation and are encouraged by your positive review of it, and we believe other valuable results will follow from this new concept.
>
> **Addressing the suggested weaknesses:**
> We would like to raise two points in response to this concern:
> 1. The stable coreset definition allows for variants that impose constraints on the median point (i.e., certain points are disallowed to be a median), such as “fair rank aggregation” that seeks a consensus ranking that satisfies certain fairness constraints (see the paragraph starting at line 057 and Experiment 3). This property does not hold for weak coresets.
> 2. The result of Danos (2021) provides a much weaker guarantee, as it requires an optimal 1-median solution over the coreset (a weak $(0,\epsilon)$-coreset), while we allow $(1+\epsilon)$-approximation as well (a stable $(\epsilon,O(\epsilon))$-coreset). Danos was unable to extend his approach to this more general condition.
>
> We thank the reviewer for all the editorial notes. They are on point and will be handled in the camera ready version.
>
> **Response to questions:**
> Yes, this component of the proof is indeed inspired by Danos. We will acknowledge this in the final version.
>
>
> Thank you for the careful reading and recognizing the aesthetic appeal of our stable coreset formulation!

---

### Official Review · Reviewer_RJA3 · 2025-10-31

**Soundness:** 4
**Presentation:** 3
**Contribution:** 2
**Rating:** 8
**Confidence:** 3

**Summary:**

The paper introduces the concept of stable coresets, which is meant to be an intermediate notion between weak and strong coresets. It is then shown that random sampling suffices to obtain a stable coreset for the 1-media problem in \ell_1 distance and in metrics that can be embedded in \ell_1. The significance of the result is that a stable coresets is a simpler proxy to strong coresets.

**Strengths:**

S1. A new notion of coresets is introduced, which interpolates between strong and weak coresets. Stable coresets captures some of the properties of strong coresets while being easy to compute via random sampling. The paper may bring some new ideas and may inspire stable coresets construction for other settings, beyond 1-median.
S2. The paper provides a rigorous analysis.
S3. Presentation is quite good, by giving insights and explaining the main contributions and results, while the proofs are given in the appendix.
S4. Improvement over state-of-the-art is quite good, where previous method gives similar error but requires examining the while dataset, while the proposed method requires only constant time per sample.

**Weaknesses:**

W1. Focusing on the simple problem of 1-median limits the practical applicability of the idea. I have the impression that the paper is a better fit for a venue in theoretical computer science.

**Questions:**

I do not have specific questions. I think that this is a good paper making a solid contribution in the topic of coresets. My only concern is whether having results only for 1-median is somewhat limited.

---

> ### Author Response · Authors · 2025-11-19
>
> We sincerely thank all reviewers for their thoughtful feedback and constructive comments. Let us first address a common point regarding our focus on $k=1$, and then respond separately to each reviewer's specific questions.
>
> Our work focuses on uniform sampling as a means to construct coresets for the median problem, providing the first sublinear-time method for important discrete metrics such as Kendall-tau and Jaccard. To enable truly uniform sampling, we restrict attention to $k=1$ because it is well-known that for larger $k$, small clusters cannot be reliably sampled. Extensions to $k>1$ via uniform sampling are possible but are more limited and somewhat cumbersome. For example, Huang et al. (2023a) restrict the dataset (the optimal clustering must be balanced) and also modify the coreset definition (to consider only balanced clusterings of the coreset $Q$). We chose to focus on $k=1$ in order to construct a stable coreset, without any such additional modifications. We stress that results do have applications (even if not for coresets) for $k>1$. Indeed, Theorem 5.4 shows an application to linear-time approximation algorithms for small $k>1$; it uses the framework of Ackermann et al. (2010), which happens to require exactly what our coresets provide (that uniform sampling works well for $k=1$).
>
> Beyond these technical considerations, the $k=1$ case is itself an aggregation task that is fundamental. It asks for a single point that represents well the entire dataset, which is important and useful by itself. In particular, it arises in several important metric spaces and was given specialized names, e.g., rank aggregation (when points are permutations), the median string problem, the genome median problem, and Wasserstein barycenter (under earthmover aka optimal transport distance). In fact, some of our experiments deal with rank aggregation.
>
> We find vast potential in the stable coreset formulation and are encouraged by your positive review of it, and we believe other valuable results will follow from this new concept.
>
> **We thank the reviewer for the positive feedback. We discussed the points raised in the discussion above.**

---

### Official Review · Reviewer_thES · 2025-11-01

**Soundness:** 3
**Presentation:** 3
**Contribution:** 3
**Rating:** 6
**Confidence:** 4

**Summary:**

The paper introduces a new variant of coreset, stable coresets, that has guarantees positioned between strong and weak which are known in literature. The work proposes instead of preserving cost for particular centers, we can preserve the relative ordering (in terms of costs) of the centers. The authors then show that for this type guarantee, uniform sampling can be used to construct a stable coreset, specifically, for the 1-median problem in $\ell^d_1$ a uniform sample of $O(\varepsilon^{-2}\log{d})$ samples is a stable coreset. They further characterize the requirement of $C-$dispersed instances, under which uniform sampling actually produces strong coresets. Experiments are provided showing the performances match coresets constructed using importance sampling.

**Strengths:**

The paper is well written and the proofs are correct to the best of my knowledge. The formalization of stable coresets is novel. The reason notion of stable coresets is useful is because the guarantees transfer to any isometrically embedded metric. This directly provides coresets (through uniform sampling) for metrics that can be embedded into $\ell_1$, the ones highlighted in this work are Jaccard and Kendall-Tau.
The characterization of $C-$dispersed instance is also nice, and may help explain the effectiveness of uniform sampling in practice.

**Weaknesses:**

To obtain the stable coreset guarantee for uniform sample, the authors first introduce the notion of relative cost difference approximation (RCDA) and argue that uniform sample of size logarithmic in VC-dimension gives a set that provides the RCDA guarantee, which is then used to prove the stable coreset guarantee.
The authors discuss how to use stable coreset for $k-$median but better results are already known in the literature.

**Questions:**

Could the authors elaborate on the structural assumptions where extensions to $k$-median is possible and how sample complexity would compare w.r.t. weak coresets?

---

> ### Author Response · Authors · 2025-11-19
>
> We sincerely thank all reviewers for their thoughtful feedback and constructive comments. Let us first address a common point regarding our focus on $k=1$, and then respond separately to each reviewer's specific questions.
>
> Our work focuses on uniform sampling as a means to construct coresets for the median problem, providing the first sublinear-time method for important discrete metrics such as Kendall-tau and Jaccard. To enable truly uniform sampling, we restrict attention to $k=1$ because it is well-known that for larger $k$, small clusters cannot be reliably sampled. Extensions to $k>1$ via uniform sampling are possible but are more limited and somewhat cumbersome. For example, Huang et al. (2023a) restrict the dataset (the optimal clustering must be balanced) and also modify the coreset definition (to consider only balanced clusterings of the coreset $Q$). We chose to focus on $k=1$ in order to construct a stable coreset, without any such additional modifications. We stress that results do have applications (even if not for coresets) for $k>1$. Indeed, Theorem 5.4 shows an application to linear-time approximation algorithms for small $k>1$; it uses the framework of Ackermann et al. (2010), which happens to require exactly what our coresets provide (that uniform sampling works well for $k=1$).
>
> Beyond these technical considerations, the $k=1$ case is itself an aggregation task that is fundamental. It asks for a single point that represents well the entire dataset, which is important and useful by itself. In particular, it arises in several important metric spaces and was given specialized names, e.g., rank aggregation (when points are permutations), the median string problem, the genome median problem, and Wasserstein barycenter (under earthmover aka optimal transport distance). In fact, some of our experiments deal with rank aggregation.
>
> We find vast potential in the stable coreset formulation and are encouraged by your positive review of it, and we believe other valuable results will follow from this new concept.
>
> **Addressing the suggested weaknesses:**
>
> Strictly speaking, the results are incomparable. We suggest a different tradeoff among the parameters $k$, $\epsilon$, and $d$, which performs better in some common regimes (since $\log d$ is usually small in practice).
>
> **Response to questions:**
> 1. The first structural assumption (cf. Huang et al., ICML 2023a) is that no optimal  cluster is much smaller than the average, namely, all clusters have size at least $\beta \cdot n/k$ for a parameter $\beta$ that affects the results. The second assumption is more intricate: in the coreset $Q$, we consider only solutions that are $\beta$-balanced (e.g., some non-balanced solution might be much better than all the best balanced ones).
> 2. To the best of our knowledge, there are no existing weak coreset results for the $k$-median problem in $\ell_1$ to directly compare with. We have not yet carried out the required analysis and prefer not to make claims without proper analysis.

---

> > ### Comment · Reviewer_thES · 2025-11-26
> > **Reponse to authors**
> >
> > I thank the authors for their response. I maintain my positive evaluation of the work.

---

### Official Review · Reviewer_tpLe · 2025-11-05

**Soundness:** 3
**Presentation:** 3
**Contribution:** 3
**Rating:** 8
**Confidence:** 4

**Summary:**

The paper proposes the notion of stable coresets which offers a trade-off between the prevalent notion of weak coresets (provide approximation guarantee only for optimal queries) and strong coreset (provide approximation guarantees for all the queries). The authors show that while the notion is much stronger than a weak coreset, a stable coreset can still be constructed using uniform sampling making it much more scalable. This is not usually the case with strong coresets. The paper shows a stable coreset for 1-median in $\mathbb{R}^d$. Experimental results demonstrate the effectiveness of stable coresets.

**Strengths:**

1) Strong coresets require finding out importance sampling distribution which is usually tough to obtain. The stable coresets give most benefits of strong coresets. So, the idea is important for scalability in general and will be of great interest to the coreset community.
2) The paper is overall well written and appears sound.
3) Experiments are done on variety of real datasets with different dimensionalities.

**Weaknesses:**

1. The paper only gives results for 1-median. Not clear if such stable coresets can be constructed for other problems.
2. The presentation of the experimental section appears slightly rushed and can be improved. For. e.g., using Uppercase for first character in title of experiment,

**Questions:**

1. Can you comment on the following: Do you think it is possible to build smaller stable coresets via importance sampling?
2. Will uniform sampling work in general for other problems?
3. For theorem 3.1 $\eta$ is independent of $c$. Why? Please explain the impact of $c$ on quality of approximation in $\mathcal{Q}$ also.

---

> ### Author Response · Authors · 2025-11-19
>
> We sincerely thank all reviewers for their thoughtful feedback and constructive comments. Let us first address a common point regarding our focus on $k=1$, and then respond separately to each reviewer's specific questions.
>
> Our work focuses on uniform sampling as a means to construct coresets for the median problem, providing the first sublinear-time method for important discrete metrics such as Kendall-tau and Jaccard. To enable truly uniform sampling, we restrict attention to $k=1$ because it is well-known that for larger $k$, small clusters cannot be reliably sampled. Extensions to $k>1$ via uniform sampling are possible but are more limited and somewhat cumbersome. For example, Huang et al. (2023a) restrict the dataset (the optimal clustering must be balanced) and also modify the coreset definition (to consider only balanced clusterings of the coreset $Q$). We chose to focus on $k=1$ in order to construct a stable coreset, without any such additional modifications. We stress that results do have applications (even if not for coresets) for $k>1$. Indeed, Theorem 5.4 shows an application to linear-time approximation algorithms for small $k>1$; it uses the framework of Ackermann et al. (2010), which happens to require exactly what our coresets provide (that uniform sampling works well for $k=1$).
> Beyond these technical considerations, the $k=1$ case is itself an aggregation task that is fundamental. It asks for a single point that represents well the entire dataset, which is important and useful by itself. In particular, it arises in several important metric spaces and was given specialized names, e.g., rank aggregation (when points are permutations), the median string problem, the genome median problem, and Wasserstein barycenter (under earthmover aka optimal transport distance). In fact, some of our experiments deal with rank aggregation.
>
> We find vast potential in the stable coreset formulation and are encouraged by your positive review of it, and we believe other valuable results will follow from this new concept.
>
>
> **Response to questions:**
>
> 1. This is an interesting direction, to consider known importance sampling constructions for strong coresets in $\ell_1$, and investigate whether weakening the guarantees from strong to stable coresets yields smaller sample sizes. While we believe that the bound we achieved via uniform sampling can be improved, this suggestion offers an alternative path to “beat” the current $O(1/\epsilon^4)$ bound for strong coresets.
>
> 2. Although uniform sampling has been extensively studied, many open problems remain, such as $k$-means in $\ell_1$ or $\ell_2$. Our work shows that the practical effectiveness of uniform sampling can be supported by rigorous theoretical guarantees. Specifically, we hypothesize that if a weak coreset can be obtained via uniform sampling for a given problem, then also a stable coreset for that problem should be constructible via uniform sampling.
>
> 3. The constant $c$ measures how well $Q$ preserves the optimal median's cost. Both parameters of the stable coreset depend on $c$. Equivalently, by setting $\epsilon' = \epsilon/c$, we obtain an $(\epsilon', 4c\cdot\epsilon')$-stable coreset, showing that larger $c$ weakens the guarantee in the second parameter.

---

### Meta-Review · Area_Chair_YKyu · 2025-12-23

**Summary:**

The paper proposes "stable coresets", a new coreset definition that preserves the relative order of candidate solution costs. The authors prove that uniform sampling yields stable coresets for the 1-median problem in $l_1$ space.

Strengths:
- Reviewers recognized stable coreset as a useful interpolation between weak and strong coresets.
- The method relies on uniform sampling, enabling sublinear time construction

Weaknesses:
- The restriction to the 1-median problem ($k=1$) is the main critiques of most reviews (tpLe, thES, RJA3, DWK3).

Overall, the paper is an interesting contribution and could be a good addition to the ICLR program

**Reviewer Concerns:**

The main limitation of the paper is that it only gives results for 1-median. the only only partially addressed such limitation.

**Reviewer Scores:**

I think the reviewers would have kept the current score

---

### Decision · Program_Chairs · 2026-01-26

Accept (Poster)